# Delaying carbon dioxide removal in the European Union puts climate targets at risk

Ángel Galán-Martín [1,2,3,7], Daniel Vázquez [1,4,7], Selene Cobo [1], Niall Mac Dowell [5,6],
José Antonio Caballero [4] & Gonzalo Guillén-Gosálbez [1✉]

Carbon dioxide removal (CDR) will be essential to meet the climate targets, so enabling its deployment at the right time will be decisive. Here, we investigate the still poorly understood implications of delaying CDR actions, focusing on integrating direct air capture and bioenergy with carbon capture and storage (DACCS and BECCS) into the European Union power mix. Under an indicative target of −50 Gt of net $CO_2$ by 2100, delayed CDR would cost an extra of 0.12−0.19 trillion EUR per year of inaction. Moreover, postponing CDR beyond mid-century would substantially reduce the removal potential to almost half (−35.60 Gt $CO_2$) due to the underused biomass and land resources and the maximum technology diffusion speed. The effective design of BECCS and DACCS systems calls for long-term planning starting from now and aligned with the evolving power systems. Our quantitative analysis of the consequences of inaction on CDR—with climate targets at risk and fair CDR contributions at stake—should help to break the current impasse and incentivize early actions worldwide.

[1] Institute for Chemical and Bioengineering, Department of Chemistry and Applied Biosciences, ETH Zürich, Vladimir-Prelog-Weg 1, 8093 Zürich, Switzerland. [2] Department of Chemical, Environmental and Materials Engineering, University of Jaén, Campus Las Lagunillas s/n, 23071 Jaén, Spain. [3] Center for Advanced Studies in Earth Sciences, Energy and Environment (CEACTEMA), University of Jaén, Campus Las Lagunillas s/n, 23071 Jaén, Spain. [4] Institute of Chemical Process Engineering, University of Alicante, PO 99, E-3080 Alicante, Spain. [5] Centre for Environmental Policy, Imperial College London, Exhibition Road, London SW7 1NA, UK. [6] Centre for Process Systems Engineering, Imperial College London, Exhibition Road, London SW7 2AZ, UK. [7] These authors contributed equally: Ángel Galán-Martín, Daniel Vázquez. ✉email: gonzalo.guillen.gosalbez@chem.ethz.ch

Due to the growing carbon emissions and rising global temperatures, carbon dioxide removal (CDR) will become essential to combat climate change[1–4]. According to the most recent integrated assessment modeling scenarios (IAMs), limiting global warming to 1.5 °C will require deploying CDR to remove 10-20 Gt/yr of $CO_2$ over the 21st century, and cutting emissions sharply to reach carbon neutrality around mid-century and ultimately become carbon-negative[2,5,6].

CDR technologies and practices deliver net negative emissions by removing and sequestering $CO_2$ from the atmosphere[7,8]. Nature-based strategies sequester the $CO_2$ in natural sinks (e.g., afforestation/reforestation, AR, and tailored agricultural practices), while engineered CDR stores the $CO_2$ either in geological sites or minerals (e.g., enhanced weathering, Bio-Energy with Carbon Capture and Storage, BECCS, and Direct Air Carbon Capture and Storage, DACCS)[8–10]. To date, most IAMs already include AR and BECCS[11], while other CDR options with low technology readiness levels and limited removal potential are often omitted[1,5,8]. BECCS is particularly appealing because it removes $CO_2$ while delivering renewable and reliable energy[12]; however, it can lead to large impacts on ecosystems and biodiversity, which could be alleviated by resorting to DACCS[13–16]. DACCS shows a large removal potential limited mainly by the storage capacity[5,17], yet its large heat and power requirements hamper its large-scale adoption[16,18,19], suggesting that an optimal regionalized portfolio of CDR strategies should be sought.

The CDR deployment to date has been minimal[20,21] with only 1.5 million t $CO_2$/yr removed via BECCS[9,22] and around 0.01 million t $CO_2$/yr[18] with DAC technologies, often deployed without long term $CO_2$ storage. The main barriers for CDR deployment include the lack of consensus on the need to start CDR today—as it is often perceived as a problem for later—, the absence of market incentives and strong political drivers, and governance challenges. Moreover, debates on the ethics surrounding CDR have also emerged over the so-called moral hazard and betting concerns about negative emissions[1,23–26]. These concerns refer to the risk of obstructing emissions cuts and delaying CDR deployment[27,28] under the assumption that CDR could be adopted to the extent and with the motion needed to compensate ongoing emissions and meet the climate goals. In practice, however, future technological, social, and environmental barriers that remain largely unexplored[29–31] may hinder the implementation of CDR and the attainment of the long-term temperature targets[26,32–35].

Deterring mitigation actions and delaying CDR is already perceived as risky[26,36], with ongoing discussions advocating the definition of separate mitigation and removal targets to promote CDR[30,36–39]. The consequences of delaying mitigation actions have already been studied[40–49], while the implications of postponing CDR remain unclear[19,36,50,51]. Hence, expanding our currently limited knowledge on the perils of CDR inaction could help break the current deadlock, expedite CDR measures, ensure ambitious contributions consistent with fair responsibilities[37] and delineate the best plan forward to combat climate change.

Here we fill this gap by studying the implications of CDR inaction, focusing on BECCS and DACCS deployment in the European Union (EU) as key engineered CDR strategies strongly linked to the energy sector[52,53]. The EU is expected to play a vital role in future CDR actions[54–56] and has proposed a legally binding climate-neutrality target by 2050[57–59] that will require CDR measures yet to be defined[56]. Applying a tailored energy systems model, we find that postponing CDR actions could increase the removal costs quite substantially (e.g., 0.12–0.19 trillion EUR2015 per year of inaction to deliver −50 Gt of net $CO_2$ emissions) and drastically reduce the removal potential, putting the EU at risk of missing targets (e.g., from −73.73 to

−35.60 Gt $CO_2$ by delaying CDR action from 2020 to 2050, respectively). Our results highlight the urgent need to deploy CDR early to meet the climate goals on time and in a cost-effective manner.

## Results

### Consequences of delayed CDR actions: extra-costs and under-exploitation of resources.
We start by analyzing the implications on costs and emissions of delaying the large-scale deployment of BECCS and DACCS. To this end, we apply an energy systems model (i.e., RAPID) that captures the interplay between the CDR technologies and the power sector to identify the minimum cost (Fig. 1a) and maximum net negative emissions potential (Fig. 1b) roadmaps, starting the CDR deployment at different points in time in 2020–2100 (optimal solutions every five years depicted with markers in Fig. 1). RAPID identifies the optimal portfolio of power technologies, BECCS and DACCS, including their location and installed capacities, which may vary over time to meet a given energy demand pattern and $CO_2$ removal target (details in Methods).

We first investigate the economic implications of delaying CDR for various net $CO_2$ emissions targets, finding that postponing CDR increases the removal cost, first smoothly and then sharply as we further postpone actions (Fig. 1a). For example, reaching carbon neutrality by 2100 by deploying BECCS and DACCS from 2020 would cost 22.8 trillion EUR2015 (in the range 18.6–26.6, uncertainty analysis in the Supplementary Tables 43–50), and 24.1 when starting in 2075 (18.2–29.5), while it would become infeasible when starting after 2080. Delaying actions would reduce the amount of cheap biomass available, making it necessary to resort to the more expensive DACCS to attain the CDR target (109–155 EUR2015 $t^{-1}$ $CO_2$ for DACCS considering optimistic prospects vs. 61–86 EUR2015 $t^{-1}$ $CO_2$ with BECCS). For example, to meet a target of −50 Gt net of $CO_2$, DACCS would have to remove 9% of the cumulative gross negative emissions needed by 2100 when starting in 2020, and up to 40% for a delayed start in 2050, after which this indicative CDR target would be unattainable. For the same CDR target, each year of CDR inaction would increase the removal cost in the range of 0.12-0.19 trillion EUR2015 by 2100. The opportunity cost of delaying CDR would vary over time, depending on the gradual mix decarbonization and improvements attained via learning curves (e.g., CAPEX of DACCS expected to become c.a. 70% cheaper in 2050 relative to 2020, Supplementary Table 14).

We next focus on the feasibility of the removal roadmaps, assuming an emergency plan to maximize CDR, starting actions in different years. We find that delaying CDR constrains the total removal potential substantially (Fig. 1b), again, first smoothly and then sharply. Notably, postponing the deployment of BECCS and DACCS beyond 2050 might prevent the EU from removing −50 Gt $CO_2$ before 2100 (i.e., EU emissions emitted in the last decade[56,60]), while delays beyond 2080 might even impede reaching carbon neutrality in the power sector. The EU would maximize its CDR potential by deploying BECCS and DACCS from today (i.e., net −73.73 Gt $CO_2$ by 2100 starting from 2020, Fig. 1b). This maximum CDR potential would be constrained by the geological storage capacity in the EU, estimated at 90.53 Gt $CO_2$ for hydrocarbon reservoirs and aquifers[61]. In practice, however, the final amount of CDR that could be delivered will be subject to social acceptance issues, regulatory limitations, competition for resources, and economic feasibility challenges. For example, deploying BECCS at scale will be challenging due to the competition for land and water with food production and other sustainability concerns, including the high demand for fertilizers required to sustain the bioenergy crops[62]. These issues

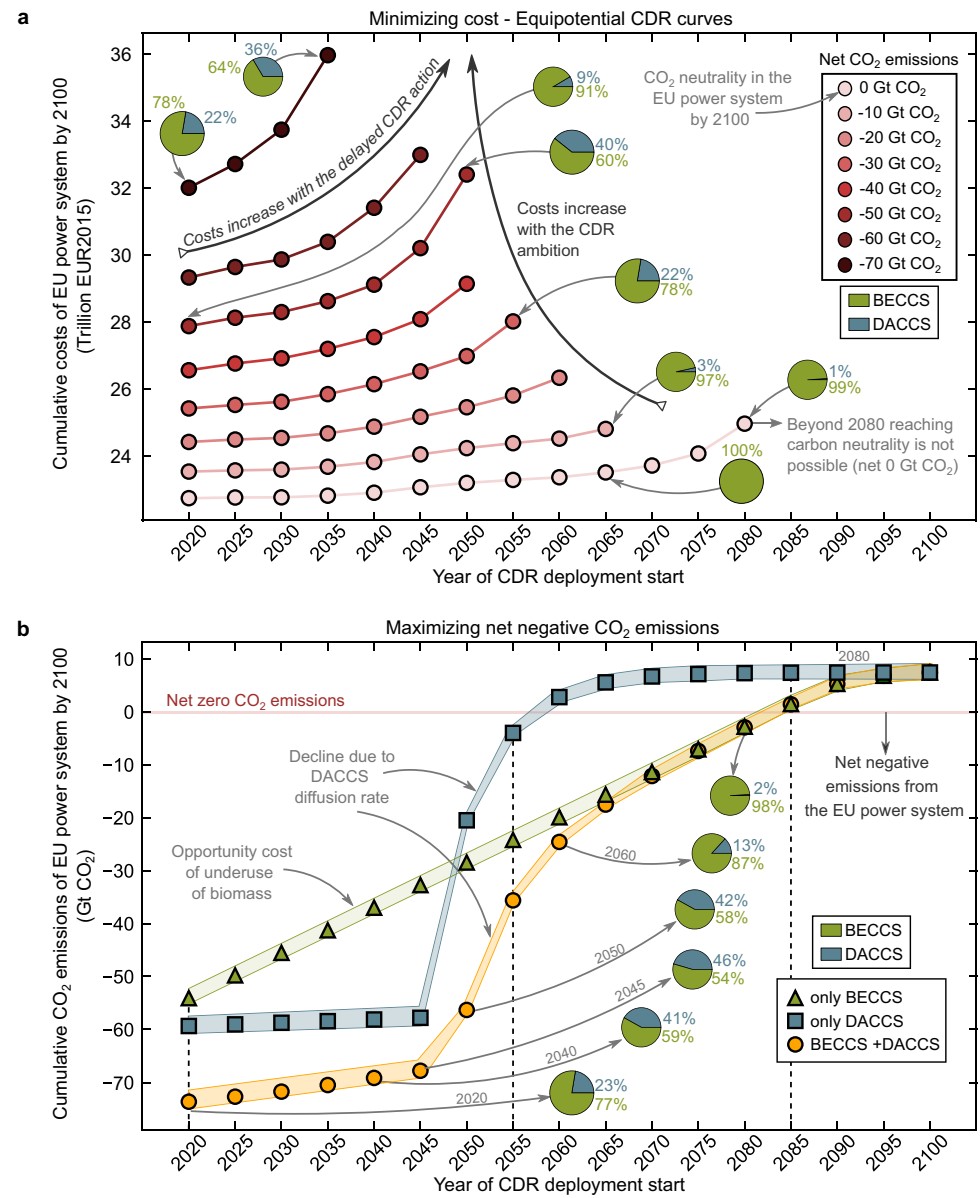

**Fig. 1 Implications on costs and emissions of delayed-actions on carbon dioxide removal (CDR) considering different starting points for bioenergy with carbon capture and storage (BECCS) and direct air carbon capture and storage (DACCS) deployment until 2100 (x-axis).** Subplot (**a**) shows the minimum costs of the European power system associated with increasing CDR targets. Subplot (**b**) shows the maximum cumulative net CDR that could be attained considering the deployment of BECCS and DACCS from a particular point in time onwards (green profile only with BECCS, blue with DACCS, and yellow considering both BECCS and DACCS). Dots correspond to the optimal solutions for the 5-year time steps starting in 2020 and ending in 2100. The shaded area in subplot (**b**) indicates the uncertainty in the life cycle $CO_2$ emissions (i.e., $\mu \pm 2\sigma$, Methods for details on the uncertainty analysis). The pie charts illustrate the proportion of gross CDR provided with BECCS and DACCS, respectively.

could be addressed (to some extent) by resorting to marginal land and residues and implementing sustainable management practises[62–64]. Hence, the biomass potentials linked to the availability of residues and marginal land are affected by uncertainties. Notably, competition for the limited biomass resources available will likely emerge due to the biomass versatility to decarbonize different sectors (e.g., transport), while marginal land availability might vary greatly due to improvements in agriculture or dietary changes[65]. We performed a sensitivity analysis to study the effects of these uncertainties, finding that reducing the biomass availability (i.e., −25% of the original estimates, Supplementary Fig. 2), would not change the maximum CDR substantially starting actions today (−70.04 Gt $CO_2$ by 2100). This is because the storage capacity would still act

as the main bottleneck. However, when CDR actions are delayed beyond 2050, the reduced availability of biomass would result in a significant drop in the maximum $CO_2$ removed (e.g., −17% and −33% starting in 2050 and 2060, respectively). The costs of the power system would also increase when biomass availability is constrained further due to the need to resort to DACCS from the early years. Note that our results only consider domestic biomass resources and onshore geological sites. However, imported biomass and $CO_2$ storage capacity beyond the EU borders and offshore storage sites may substantially increase the potential despite facing international governance issues while posing sustainability and social justice questions[66,67]. Hence, considering other $CO_2$ storage options (e.g., mineral trapping) or other strategies beyond the power sector (e.g., planting trees or

improving soil carbon sequestration) could further increase the EU's CDR ambition. For example, leaving aside the shortcomings related to permanence and vulnerability of the carbon sequestration in trees, reforesting the EU countries could provide an additional removal potential of 0.91 Gt $CO_2$ per year until the sink saturates (i.e., 30 years when the forest reaches the steady-state). Similarly, improved management practices such as delaying harvests or adopting reduced-impact logging would further remove 0.07 Gt $CO_2$ per year by 2100[37,68]. The gradual temporal decline in CDR potential is due to the underused resources (i.e., biomass residues and unexploited marginal land to grow energy crops) and the maximum diffusion rates of the CDR technologies. Lacking sufficient BECCS capacity to process the biomass residues, they would degrade and eventually release the biogenic carbon to the air in the form of $CO_2$, while the unused marginal land would represent an opportunity lost. Furthermore, the technology diffusion factor constraining the deployment speed (e.g., 20% of annual capacity up-scaling[69]) critically limits the maximum attainable DACCS capacity. The potential reduction varies linearly over time for BECCS (green curve in Fig. 1b) and follows a sigmoidal shape for DACCS, with a critical point around 2045. This behavior is due to the unused biomass and land resources, the main factors constraining BECCS, which accumulate almost linearly over time. In contrast, the maximum attainable DACCS capacity, only limited by the diffusion rate, increases exponentially over time; consequently, further delays result in much fewer DACCS plants ready to be deployed on time.

Overall, in both the cost-optimal and the maximum removal roadmap, BECCS emerges as predominant regardless of the starting year, providing double benefits by contributing to CDR while delivering reliable power to meet the demand (CDR contribution breakdown in pie charts in Fig. 1a, b). When maximizing the CDR potential, the DACCS contribution increases with later starting years, while the BECCS contribution declines due to the loss of biomass potential (e.g., DACCS from 23% starting in 2020 to 46% in 2045, Fig. 1b for the yellow profile). However, the initial capacity for DACCS of 1 Mt of $CO_2$ captured per year—reflecting the current scale ambition—and the maximum growth rate of 20%/yr observed in the historical deployment of power technologies[19,51,69] would strongly limit the maximum DACCS capacity when actions are further delayed. Hence, deploying DACCS from 2050 would result in a contribution of −30.57 Gt of $CO_2$ by 2100 (i.e., 42% of the total gross −72.94 Gt of $CO_2$ removed), while the DACCS share starting after 2080 would become negligible (i.e., <2%). In contrast, the BECCS initial capacity set to 250 MW—reflecting the state-of-the-art largest biomass-fired power plants—results in the deployment pace being limited mainly by the geological capacity, except when starting actions near 2100, where the diffusion rate constraint becomes the bottleneck. Notably, the synergetic integration of BECCS with DACCS into the energy system emerges as an appealing option to enhance the $CO_2$ removal capacity (yellow curve showing always higher net $CO_2$ removal than the green and blue profiles). Therefore, deploying DACCS appears as a complementary option under a rapid emergency deployment, lacking biomass resources or if BECCS deployment is constrained by the diffusion speed, being environmentally benefited from the carbon-negative electricity supplied by BECCS at the expense of increasing costs[19,52,70].

Our findings are particularly relevant in the context of the EU Green Deal aiming at climate neutrality by 2050. The EU Climate Law fails to explicitly discuss the role of CDR technologies (other than land sink removals) to meet such a goal. However, to become climate neutral in 2050 (and provide negative emissions beyond then), it seems clear that some countries and sectors will

rely on CDR to offset emissions. In this context, the EU power sector will likely play a key role in meeting the climate neutrality target by reaching net-zero emissions before 2050 and then becoming carbon negative to compensate emissions from hard-to-abate sectors. Considering the 2020 to 2050 horizon, delaying the deployment of BECCS and DACCS beyond 2040 might prevent the EU power sector from reaching carbon neutrality in 2050 (e.g., −0.85 Gt $CO_2$ by 2050 starting in 2040) while increasing the costs in the range of 0.04–0.10 trillion EUR per year of inaction (Supplementary Fig. 3). Therefore, promoting the near-term integration of those technologies at the earliest is vital to ensure that they can be deployed to the extent required to meet the long-term goals.

**Emissions pathways through 2100 for delayed CDR actions.** We next analyze the emission pathways, considering three representative scenarios, namely, NOW, SLOW, and LATE, which maximize the removal potential starting CDR actions in years 2020, 2055, and 2085, respectively (Methods section). The net $CO_2$ emissions balance accounts for the: i) gross removal, i.e., total $CO_2$ removed from the air, either via photosynthesis (BECCS) or through physicochemical processes (DACCS); ii) anthropogenic life cycle emissions embodied in the power technologies, BECCS and DACCS supply chains; and, iii) $CO_2$ emissions during biomass/natural gas combustion in BECCS/DACCS facilities, respectively, due to capture efficiencies below 100%. These emissions contributions vary across pathways, which differ in the availability of resources and maximum BECCS and DACCS capacities (constrained by the technology diffusion rates).

In the NOW scenario (Fig. 2a), starting CDR actions in 2020, the gross negative emissions would amount to −94.05 Gt $CO_2$ by 2100, mostly provided by BECCS (77% in absolute terms) and a small contribution by DACCS (23%), yielding −73.73 Gt of net $CO_2$ removed from the atmosphere. This scenario shows the largest positive residual emissions (+20.31 Gt $CO_2$), 63% attributed to the uncaptured biogenic $CO_2$ and the life cycle emissions linked to BECCS supply chains (+7.11 and +5.74 Gt $CO_2$, respectively), 30% due to the life cycle emissions of the other power technologies (excluding BECCS), 5% associated with the life cycle emissions from DACCS and, finally, a marginal contribution from the $CO_2$ transportation and storage infrastructure (<+1%). This scenario would fully exhaust the domestic storage capacity in the EU, which would act as a bottleneck for CDR. Furthermore, almost all the biomass resources available would be exploited (i.e., 95% of the biomass residues and 94% of marginal land), with a small amount lost due to the limited rate at which BECCS could be scaled up during the first years (Supplementary Fig. 4). The overall storage efficiency —i.e., total net $CO_2$ removed per kg of $CO_2$ stored— would reach 81%, where most geological sites would store the biogenic $CO_2$ captured via BECCS (71%), a smaller amount of atmospheric $CO_2$ captured with DACCS (24%), and finally the captured emissions linked to the heating needs of DACCS (5%). Notably, when starting CDR actions today, the geological capacity needed to store the $CO_2$ captured in fossil power plants with CCS would be negligible, thus delivering the maximum CDR constrained by the domestic storage capacity (Fig. 3). DACCS would play a role in complementing BECCS and ultimately helping to remove $CO_2$ at the pace required, benefiting from the carbon-negative electricity delivered in the system, and exploiting its flexibility to be located closer to the geological sites in countries with scarce biomass resources[52,70].

In the SLOW scenario starting CDR actions in 2055, the maximum gross and net removal potential would drop considerably (−49.61 Gt and −35.60 Gt, respectively, vs.

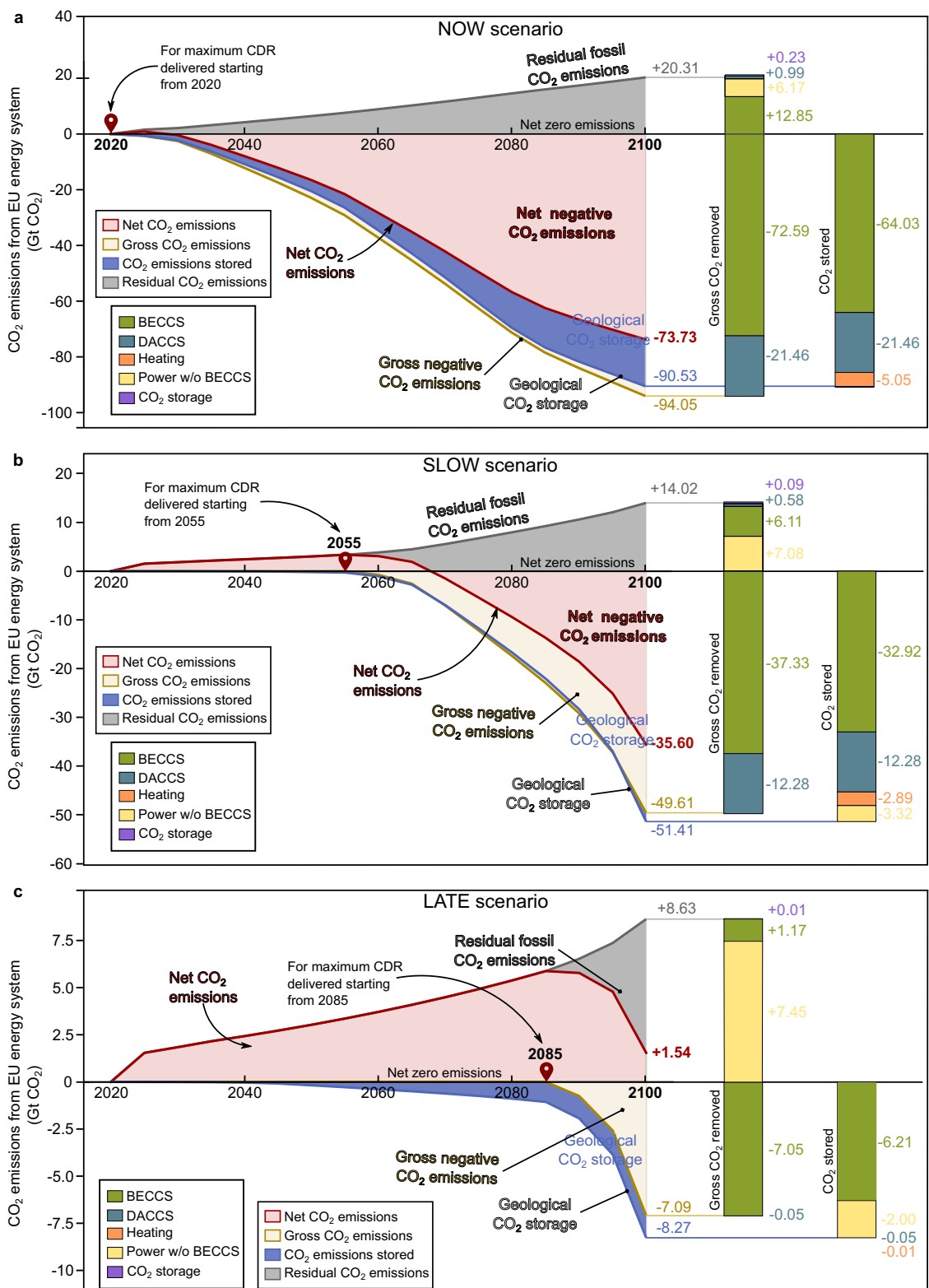

**Fig. 2 CO₂ emission pathways and breakdown in the European energy sector considering the three illustrative scenarios with different starting points for the deployment of bioenergy with carbon capture and storage (BECCS) and direct air carbon capture and storage (DACCS) by 2100.** Subplot (**a**) corresponds to the NOW scenario starting carbon dioxide removal (CDR) in 2020. Subplot (**b**) corresponds to the SLOW scenario starting CDR in 2055. Subplot (**c**) corresponds to the LATE scenario starting CDR in 2085. Stacked bars on the right show the breakdown by the source of the life cycle residual emissions, gross negative emissions, and stored emissions.

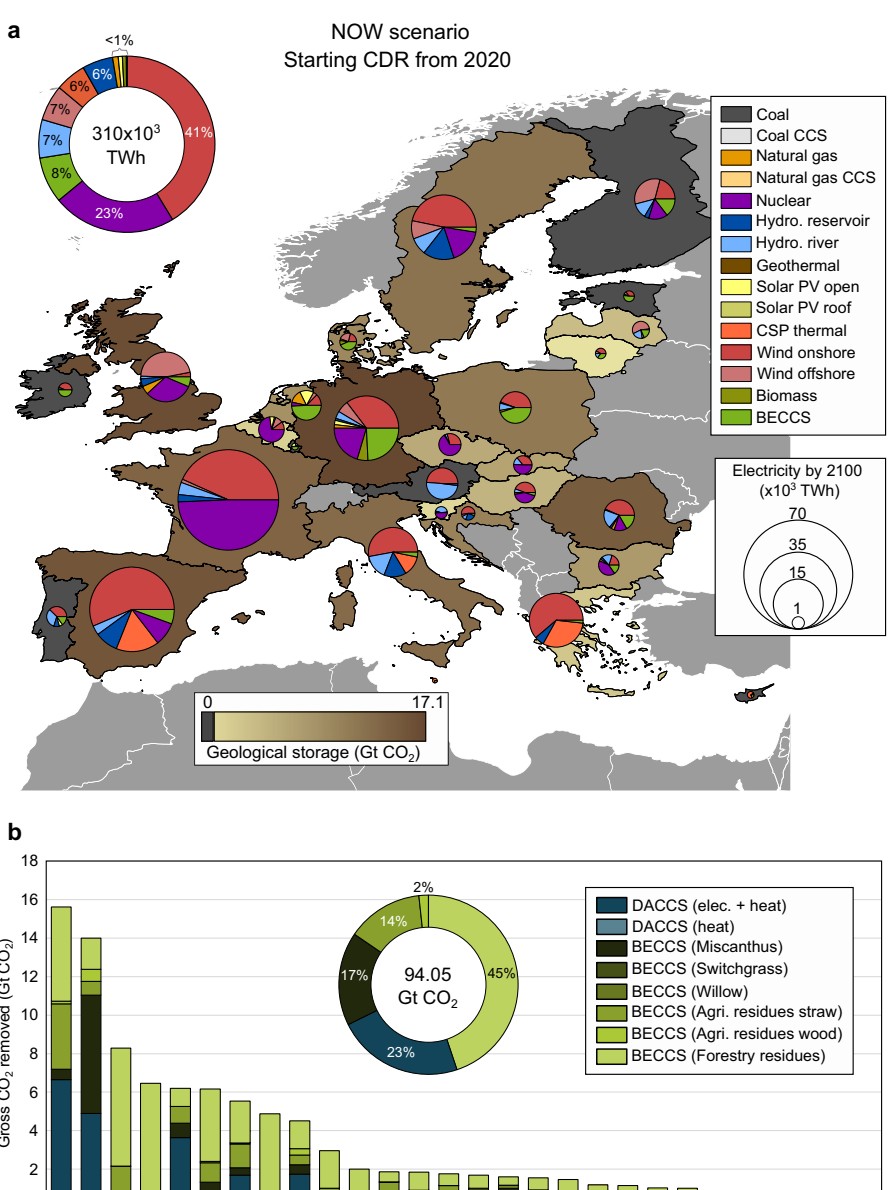

**Fig. 3 Regional implications for the European energy system starting the deployment of bioenergy with carbon capture and storage (BECCS) and direct air carbon capture and storage (DACCS) in 2020 (NOW scenario).** Subplot (**a**) corresponds to the optimal electricity generation by 2100 in each European country. The pie charts show the share of generation per electricity technology depicted with different colors, while the size of the pie charts is proportional to the generation by 2100 (TWh). CCS, PV and CSP stand for carbon capture and storage, solar photovoltaic and concentrated solar power, respectively. Each country is colored according to the CO$_2$ stored in the geological sites; the darker the shade, the greater the CO$_2$ stored. Subplot (**b**) shows the breakdown by country of the gross CO$_2$ removed from the atmosphere considering the different biomass resources for BECCS and DACCS technologies. Countries in subplot (**b**) are labeled according to the ISO3 code abbreviation. The map in subplot (**a**) was created using ArcGIS® 10.7.1 software by Esri[82]; no copyrighted material was used.

−94.05 Gt and −73.73 Gt in the NOW scenario), while the storage capacity would not be fully exhausted (only 57% of the capacity utilized). Notably, net negative CO$_2$ emissions would not be achieved until 2070 due to the need to offset the residual emissions taking place until that year. Before 2080, the deployment rate would limit the BECCS removal capacity, while beyond 2080, biomass resources would become the bottleneck (residues and land). 86% of the residues and 90% of the marginal land available from 2055 to 2100 would be exploited, representing only 63% and 57% of their respective total potentials (if actions were started in 2020 and continued until 2100). In contrast, the

maximum deployment rate would constrain the DACCS capacity (Supplementary Fig. 4). Furthermore, the storage efficiency would be reduced to 69%, with 88% of the storage devoted to atmospheric carbon and the remaining part storing fossil carbon (i.e., from natural gas combustion powering DACCS and power plants with CCS). The global EU geological storage capacity would not be fully depleted[5], yet competition between fossil and atmospheric captured CO$_2$ for the sites available at the regional level could become an issue, particularly given the increasing policy support for CCS in fossil-fired power plants in many countries[71].

Finally, in the LATE scenario, the maximum gross negative emissions would be substantially reduced to $-7.09$ Gt $CO_2$. The removal potential would be limited by the maximum diffusion rates of BECCS and DACCS, which would even impede reaching $CO_2$ neutrality in the EU power sector ($+1.54$ Gt of net $CO_2$ emissions by 2100) and constrain the use of residues and land to 40% and 20% of their maximum availability from 2020 to 2100, respectively. Most of the biomass resources would be consumed by biomass power without CCS during the inaction periods (only 8% of the total residues and 14% of the land available in the period 2020–2100 would be exploited with BECCS after 2085). Here, DACCS would play a minor role, removing only $-0.05$ Gt of $CO_2$ by 2100 due to the slow speed at which it could be scaled up. The storage efficiency would substantially decrease, with 25% of the total capacity available devoted to fossil $CO_2$ emissions.

Overall, delaying the CDR deployment would lead to the underuse of biomass and land resources, tighter bounds on the BECCS and DACCS facilities, and domestic storage sites depleted with fossil carbon, which altogether would reduce the future ability of individual countries on CDR. However, transboundary agreements enabling import/exports of biomass and $CO_2$, new estimates of suitable geological sites, less conservative biomass potentials, and a broader CDR portfolio beyond the energy sector could enhance the EU's ability to deliver net negative emissions[37].

**Regional implications for the energy systems**. The carbon-negative electricity supplied by BECCS and the large electricity and heating requirements of DACCS create strong links between them and the power system. Therefore, their deployment would require long-term planning, ensuring their effective integration into the evolving portfolio of power technologies starting at the earliest. In the NOW scenario (shown in Fig. 3, SLOW and LATE scenarios in Supplementary Figs. 5 and 6, respectively), the total electricity generated in the EU by 2100 would be produced mostly by wind onshore (41%), followed by nuclear (23%), BECCS (8%), hydropower run-of-river and wind offshore (both with 7%), concentrated solar power (6%), hydropower from reservoirs (6%), and marginal contributions from natural gas, solar photovoltaic open ground, biomass w/o CCS, and geothermal (<1%) (Fig. 3a). Notably, BECCS becomes relevant in the generation portfolio, providing firm capacity and ancillary services to support the high penetration of intermittent technologies with dispatchable carbon-negative electricity. Overall, a handful of countries would shoulder most of the CDR efforts. Only four countries would deliver almost half of the gross removal by 2100, with France and Spain at the top deploying both BECCS and DACCS, followed by Germany and Sweden deploying only BECCS (i.e., 44.37 Gt out of 94.05 Gt of gross $CO_2$ removed, Fig. 3b).

Most of the BECCS capacity would be installed in Germany, Poland, the Netherlands, Spain, and Finland (69% of the total), exploiting their abundant biomass resources and also taking advantage of their central location (in the case of Germany and Poland). Spain, France, Germany, Sweden, and Poland would provide most of the biomass resources, i.e., 54% of the total gross $CO_2$ removed via BECCS ($-38.99$ out of $-72.59$ Gt of $CO_2$ removed with BECCS, Fig. 3b). Both forestry and agricultural residues would be fully exploited in all countries starting BECCS deployment from today. Forestry residues would contribute the most to the $CO_2$ removal (i.e., 45% of the total gross $CO_2$ removed by 2100), while miscanthus production would occupy all the marginal land available due to its overall superior carbon sequestration potential, removing $-15.87$ Gt $CO_2$ by 2100 (17% of the total gross removed) and becoming the main carbon sink in some countries (i.e., $-6.16$ Gt $CO_2$ removed in Spain). Switch-grass would also be cultivated in some regions, and the storage

capacity would no longer be the bottleneck if actions were further delayed (SLOW and LATE scenarios in Supplementary Figs. 5b and 6b, respectively). This is due to its relatively higher carbon removal capacity (i.e., $CO_2$ uptake per kg of pellets), which provides more CDR in the remaining time but at the expense of reducing the electricity delivered owing to its lower energy density.

Regarding DACCS, the configuration relying on electricity and heating would be the only one installed, benefiting from the decarbonized electricity mix. In the NOW scenario, DACCS would be established in eleven countries, with France, Spain, the United Kingdom, Italy, and Romania providing 97% of the gross removal from DACCS (i.e., $-18,72$ out of the $-21.46$ Gt $CO_2$ by 2100), all of them with enough geological sites for storing the captured $CO_2$ domestically (Fig. 3b). For example, in France, $-6.65$ Gt $CO_2$ would be removed via DACCS by 2100, taking advantage of its abundant saline aquifers and decarbonized mix dominated by wind onshore and nuclear (Fig. 3a). Similarly, in the United Kingdom, which lacks enough biomass resources to exploit its storage capacity only with BECCS, $-3.64$ Gt $CO_2$ would be removed with DACCS and stored in domestic geological sites. In practice, this roadmap would require a substantial number of DACCS facilities across the EU, i.e., around 268, with a capacity of 1 Mt $CO_2$/yr (i.e., the largest DAC plant under development today), out of which 83 would be installed in France, 61 in Spain and 46 in the United Kingdom. These DACCS plants would make the said countries incur extra costs and suffer adverse environmental impacts, such as those linked to the land requirements of the air contactors and the energy technologies powering them. Delaying actions until 2055 (SLOW scenario) would imply that less biomass is available, so the BECCS capacity would diminish accordingly, and additional DACCS facilities would be deployed in Czech Republic, Denmark, and Slovakia to maximize CDR, taking advantage of the geological storage available (Supplementary Fig. 5).

Wind capacity would be massively deployed in most countries, becoming the dominant source in France, Spain, Italy, Germany, United Kingdom, Sweden, and Finland (NOW scenario in Fig. 3a). Notably, offshore wind would become predominant in some countries, with 82% of the EU capacity located in the United Kingdom, Finland, Germany, and Sweden. Solar power plants would be deployed mostly in southern locations (i.e., Spain, Italy, and Greece), with substantial capacities of concentrating solar thermal power installed in high irradiation areas providing dispatchable renewable electricity. Some countries would also exploit their hydropower capacities, such as Austria, France, Italy, Sweden, and Spain. The contribution of nuclear by 2100 would fall in the range 0.28-35.84 TWh, with France showing the largest shares and the Netherlands the lowest. No single new nuclear plant would be installed because we assume that the capacity of this technology cannot be expanded. The existing coal plants would be completely phased out. Natural gas plants w/o CCS would still be required in United Kingdom, Germany, the Netherlands, Romania, and the islands (Ireland, Malta, and Cyprus), while playing a marginal role in the others. Overall, the integrated power-CDR system would result in carbon-negative electricity due to the high penetration of BECCS in the EU power system (i.e., $-0.24$ t $CO_2$/MWh), yet it would unavoidably increase the Levelized Cost of Electricity (LCOE) to 113.03 €/MWh.

The optimal roadmap, assuming full cooperation among EU countries, would entail an intensive trade of biomass, $CO_2$, and electricity (Fig. 4 for the NOW scenario and SLOW and LATE scenarios in Supplementary Figs. 7 and 8). However, to minimize the transport flows, the vast majority of the biomass demand would still be supplied domestically, the captured $CO_2$ stored

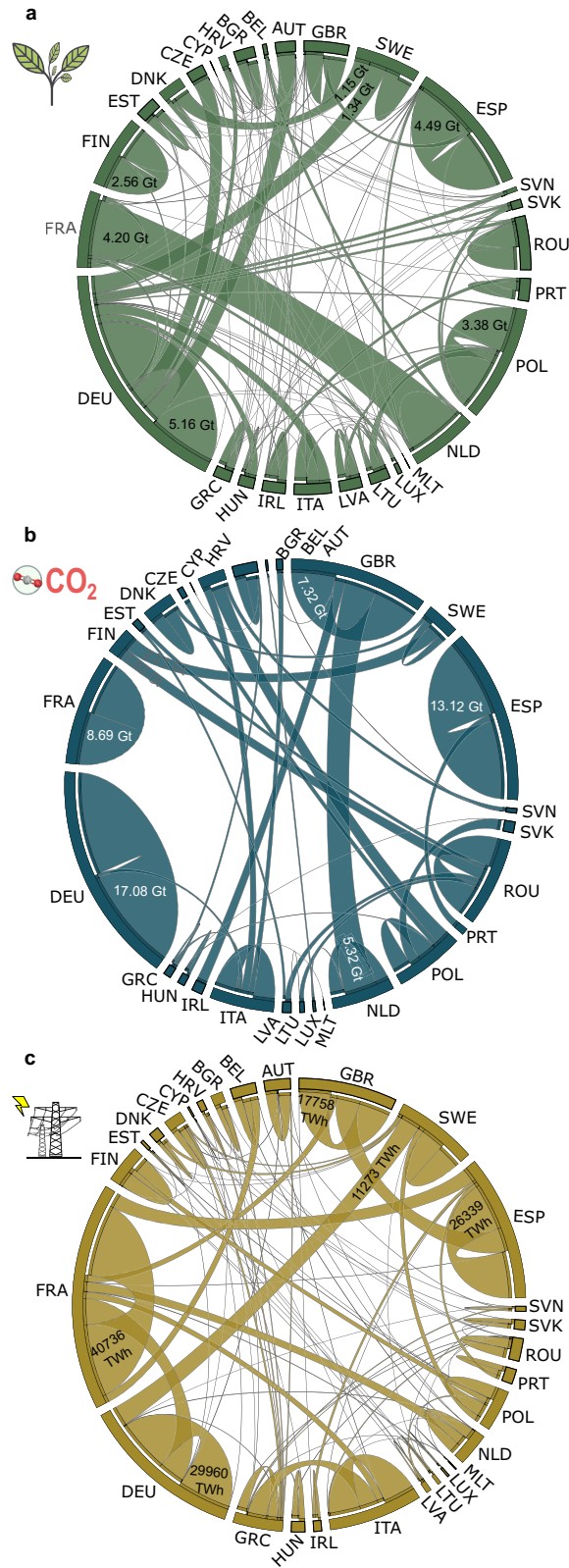

**Fig. 4 Biomass trade, $CO_2$ flows, and electricity transmission in the NOW scenario by 2100.** Subplot **a** shows the biomass traded in the form of pellets between European countries. Subplot (**b**) shows the $CO_2$ transported via a pipeline between European countries. Subplot **c** shows the electricity traded between European countries. In the chord diagrams produced using Circos[83], the European countries are depicted by arcs on the outer part of the circular layout, where the arc length provides the total biomass (subplot **a**), $CO_2$ (subplot **b**), and electricity (subplot **c**) imported, exported and consumed/stored domestically (the latter refers to chords leaving and entering the same country). Each chord represents a flow, where its thickness is proportional to the magnitude of the trade (some values are indicated for illustrative purposes). Chords directly connected to the countries' arcs represent an export (i.e., exporter country) while those non-connected (separated by a white layer) correspond to imports. Countries are labeled according to the ISO3 code abbreviation.

Finland, or Portugal becoming net exporters. Regarding electricity trade, countries such as France, Spain, and Sweden would emerge as pivotal in the power system, acting as net exporters of electricity to exploit their abundant low-carbon intensity resources (e.g., electricity trades from France to Germany, Italy, Netherlands, Belgium, and the United Kingdom, Fig. 4c).

The largest exchanges of biomass and $CO_2$ would occur between France-the Netherlands, and the Netherlands-the United Kingdom, respectively (Fig. 4a, b). Notably, Sweden would export biomass resources to Germany and Denmark to fully exploit its abundant forestry residues (i.e., 1.34 and 1.15 Gt of pellets on a dry basis, respectively, Fig. 4a). Other countries would export $CO_2$, e.g., Finland would send 2.21 Gt $CO_2$ via pipeline to the abundant deep saline aquifers and hydrocarbon fields in Sweden ($CO_2$ trades in Fig. 4b). Some countries would be almost self-sufficient in terms of biomass resources, like Portugal, which would transport $CO_2$ to the Spanish geological sites due to its low geological capacity. Overall, the transport of electricity (and $CO_2$) would be prioritized over the transport of biomass due to the larger emissions of the latter considering a given electricity demand (e.g., 0.01 vs. 0.11 kg $CO_2$ to satisfy one kWh in an importing country, respectively, considering miscanthus as biomass source and a distance of 800 km in both cases). Hence, BECCS plants would be mostly installed near the biomass sources, leading to decentralized supply chains spread across the EU territory. However, some countries would still import biomass pellets to reduce their reliance on energy from outside the country's borders and to support the high penetration of intermittent wind and solar with dispatchable carbon-negative electricity from BECCS (e.g., the Netherlands importing pellets from France or Denmark from Sweden, Fig. 4a).

The uneven distribution of domestic capacities (i.e., biomass resources, storage sites, and renewable resources) would make national and transnational collaboration essential to exploit bio-geophysical endowments[67,72] and remove $CO_2$ to the extent required. Hence, new agreements and regulatory frameworks will be needed, and shaping them may further delay CDR deployment.

## Discussion
Here we studied the implications of delaying the roll-out of CDR to raise concerns on the need to set effective plans to promote its large-scale deployment at the right time to avoid extra costs and miss climate targets.

To shed light on the economic, environmental, and technical implications of the prolonged delay of CDR actions, we focused on the deployment of BECCS and DACCS in the EU as

locally, and the electricity generated consumed on-site (domestic consumption in Fig. 4 depicted with the chords leaving and entering the same country). Some regions would be net exporters of biomass (e.g., France or Sweden) and some net importers (e.g., Netherland, Germany, or Denmark). The same would apply to $CO_2$, with, for example, Sweden, the United Kingdom, and Romania acting as net importers and the Netherlands, Poland,

prominent strategies intrinsically linked to the power system. We found that postponing CDR could substantially increase the total cost of the power system, with each year of inaction translating into 0.12–0.19 trillion EUR2015 of extra cost to meet the -50 Gt of net $CO_2$ target. Notably, this extra cost, which could be avoided in the EU by starting actions from today, compares to the estimates of the additional annual investments required globally by mid-century in the energy systems to limit global warming to 1.5 °C (i.e., between 0.15–1.70 trillion USD2010[2]). Moreover, delaying CDR actions would critically limit the removal potential (e.g., −73.73 Gt $CO_2$ starting in 2020 and −35.60 Gt $CO_2$ after 2055) due to the underuse of biomass residues and marginal land and the maximum diffusion rates of BECCS and DACCS. Hence, postponing CDR deployment beyond mid-century might prevent the EU from delivering a CDR level aligned with its fair responsibility and its expected leading role[37,56]. After 2050, the maximum CDR potential would be reduced to −56.35 Gt $CO_2$, representing around the $CO_2$ emitted by the EU only during the last decade and less than 10% of the global CDR required in pathways showing limited overshoot of the 1.5 °C target[2]. With the EU contributing short and its leadership efforts questioned, there might be a row of dominoes with other countries remaining impasse on CDR, which ultimately may impede a fair and rapid transition for effective climate change mitigation.

Furthermore, we found that delaying BECCS and DACCS may result in the inefficient use of the available domestic geological storage. Retrofitting the existing coal and gas-fired power plants with CCS to decarbonize the power sector would limit the storage capacity available, potentially raising competition issues with atmospheric $CO_2$ sequestration. Although the global geological capacity is deemed sufficient to meet the climate targets, it could become a bottleneck at the regional level due to the asymmetric distribution of storage sites. With policies promoting CCS in fossil fuel power plants, some countries might exhaust the most suitable geological sites with fossil $CO_2$, which ultimately will compromise their future CDR commitments. This reinforces the need to phase out coal and natural gas for a full transition towards renewable energy systems. Nevertheless, these policies could offer a testbed to support CCS projects, including the development of the $CO_2$ transportation and storage infrastructure needed to support CDR schemes. Without a supportive policy framework, CDR will never take off in the short term despite being imperative to offset hard to avoid energy and non-energy emissions. In this context, governance issues should also be considered to ensure that BECCS and DACCS will remove $CO_2$ at the scale and pace required. BECCS and DACCS supply chains involving multiple countries and stakeholders will encompass a wide range of activities (e.g., biomass growth, storage, transportation and processing, and $CO_2$ transportation and storage). In the absence of well-designed policy mechanisms rewarding all CDR actors or direct incentives to engage with BECCS and DACCS, their large-scale deployment is unlikely to occur. Hence, integrating BECCS and DACCS into the evolving power systems must be done at the earliest, ensuring the best use of the existing resources through effective long-term strategic decisions and planning.

CDR is gaining increasing attention in the political agendas of governments, stakeholders, businesses, and industry. The recent EU climate neutrality target by mid-century puts CDR back on the table of reticent countries[59]. Hence, this study provides valuable insight for CDR policymaking and future roadmaps aiming to integrate BECCS and DACCS value chains into the EU power systems. At the regional level, the least-cost distribution of efforts may differ greatly from equitable shares and the willingness to act in different countries[37,73]. In practice, key countries might still be reluctant concerning CDR. These nations might not fully participate and fail to shoulder their fair share of the EU CDR burden, which will result in some countries having to do more and faster and others contributing less and going slowly towards the CDR required. Nevertheless, our findings demonstrate that failure to start CDR at the right time would increase the total costs and make climate targets slipping out of reach.

Overall, our work underscores the importance of taking early actions on CDR. The potential economic and environmental benefits that would be missed if we delayed action may act as an incentive to spur CDR efforts worldwide. We focused on BECCS and DACCS and the EU, yet other technologies should also be considered and evaluated at the regional and global levels to further motivate early CDR deployment. Regardless of the technological and spatial scope of the analysis, the underuse of resources and the diffusion rates will limit our ability to remove atmospheric carbon and increase the associated costs. We hope that this quantitative analysis will help to break the current climate impasse, accelerate the take-off of CDR and get CDR deployed at the right time to avoid missing the climate targets.

## Methods

**The RAPID model**. We developed a bottom-up multi-period linear programming model to explore the consequences of delaying CDR actions, referred to as the RAPID model (RemovAl oPtImization moDel). RAPID is an energy systems model that identifies the most cost-effective emissions and technology pathways by jointly optimizing the power mix and the deployment of BECCS and DACCS, where CDR is assumed to start from a particular year within the time horizon. RAPID can be solved in two alternative ways, i.e., by minimizing the power system's costs to meet a net CDR target or by maximizing the net negative emissions balance. RAPID includes a set of technical, cost, and emissions-related constraints that the solution sought should satisfy. In each run, the model can retrofit the power mix from 2020. However, it can only deploy BECCS and DACCS from a given year defined beforehand and varied iteratively across runs. RAPID is implemented assuming perfect foresight for the modeling timeframe, i.e., the model optimizes decisions for every period with full visibility of the entire time horizon. For simplicity, the model considers five-year intervals between 2020 to 2100. Note that RAPID is flexible, and both the duration of the intervals and the horizon could be modified depending on the needs.

RAPID is mathematically formulated in compact form as follows.

$$\min_x TC = \sum_{i \in I} \sum_{j \in J} \sum_{t \in T} C_{ijt} x_{ijt} \qquad (1)$$

$$\min_x NE = \sum_{i \in I} \sum_{j \in J} \sum_{t \in T} E_{ijt} x_{ijt} \qquad (2)$$

$$A_{ijt} x_{ijt} \le W_{ijt} \forall i, j, t \qquad (3)$$

$$x_{ijt} \in \mathbb{R} \qquad (4)$$

where continuous variables $x_{ijt}$ denote technical decisions, e.g., the area of land devoted to a dedicated energy crop, the amount of residues exploited, the technologies' capacities, the transportation flows between EU countries (of biomass resources and $CO_2$), and the amount of electricity generated. These decisions are optimized for every country $j$ and time period $t$, considering a set of technologies $i$. The model can be solved by minimizing the total cost (TC) as in Eq. 1, for a given CDR target, or by minimizing the net emissions (NE), as in Eq. 2. $C_{ijt}$ are cost parameters for each technology $i$ in each region $j$ and year $t$, while $E_{ijt}$ are emission coefficients for each technology $i$ in each region $j$ and period $t$. Net $CO_2$ targets are here imposed jointly on all countries by assuming a cooperative strategy, i.e., $\sum_{i \in I} \sum_{j \in J} \sum_{t \in T} E_{ijt} x_{ijt} \le \alpha$ where $\alpha$ denotes the joint CDR target. Equation 3 represents technical constraints (e.g., limits to the penetration of intermittent technologies, demand satisfaction constraints, and technology diffusion constraints) as well as equations that quantify the economic and CDR potential (Supplementary Information). Parameters $A_{ijt}$ denote cost and emissions data as well as technological parameters, while $W_{ijt}$ are parameters appearing on the right-hand side of the constraints.

RAPID covers 15 state-of-the-art power technologies and the most prominent CDR engineered options, including (i) conventional fossil power generation technologies and their retrofit with CCS systems (coal and natural gas); (ii) firm clean technologies, such as nuclear, biomass, hydropower, and geothermal; (iii) intermittent technologies, such as wind onshore, wind offshore, solar photovoltaic open-ground and solar photovoltaic flat roof installation; and (iv) CDR technologies, i.e., BECCS based on six different types of biomass resources and two DACCS technologies. RAPID models the entire supply chain of BECCS, i.e., from the cultivation, harvesting, and pelletizing of the biomass resources to road

transportation and conversion in power plants with CCS systems. The model considers that biomass-based power technologies compete for the second-generation biomass resources available, including high-productivity energy crops grown on marginal land (i.e., short-rotation woody willow and two perennial grass sources, miscanthus, and switchgrass) and residues from agriculture and forestry activities. Regarding DACCS, RAPID includes proven technology based on high-temperature aqueous sorbents, fully powered by natural gas or with both electricity and heating from natural gas. The model considers the transportation of $CO_2$ via pipelines and its injection in geological sites (i.e., deep saline formations, depleted hydrocarbon fields, and coal fields). For all these technologies, the model considers exogenous learning costs curves as well as realistic diffusion rates limiting their deployment.

RAPID optimizes decisions at the country level taken from 2020 to 2100 (five-year periods) in the EU, divided into 28 countries. These temporal and spatial scales are consistent with the scope of the analysis. Each country is modeled as a load node with specific temporal patterns of demand and resource availability. Distances between countries are quantified based on their centroids. Domestic transport of biomass and $CO_2$ within countries considers a standard distance of 100 km. Note that BECCS and DACCS supply chains may span several countries, e.g., miscanthus cultivated in country A, transported to country B in the form of pellets to be combusted, and the captured $CO_2$ transported by pipeline and injected in a saline aquifer in country C.

RAPID was implemented in the General Algebraic Modelling System (GAMS) software[74] version 32.2.0 and solved with the CPLEX solver on an Intel i9-9900 CPU, 3.10 GHz computer with 32 GB RAM. RAPID features 305,314 continuous variables and 109,068 equations; the solution time is always in the range 10-120 min depending on the instance solved. The computer code supporting our analysis is available from the corresponding author upon request. A detailed mathematical description of the RAPID model and the underlying assumptions can be found in Supplementary Information Section 1 on "The RAPID model" (Supplementary Equations 1-53). Data inputs are described and presented in Supplementary data (Supplementary Tables 5–42).

**Scenario definition and solution approach**. We define three different scenarios, labelled as NOW, SLOW and LATE, differing in the starting year for the CDR deployment. The NOW scenario considers immediate CDR action starting in 2020. The SLOW scenario assumes that CDR action starts in 2055, while the LATE scenario delays CDR until 2085.

We solve RAPID considering the time horizon 2020-2100, divided into 16 periods $t$ of five years length each, assuming that BECCS and DACCS can only be deployed from a particular period $t'$ onwards. For example, when RAPID is solved for 2050, we assume that CDR can only be deployed from 2050 to 2100, but not before. Therefore, not deploying BECCS during inaction periods implies that biomass residues are not mobilized, and the marginal land remains unexploited, resulting in a CDR potential loss. In contrast, changes in the power mix can occur at any year within the time horizon, assuming perfect foresight modeling. Power plants and BECCS and DACCS facilities installed in period $t$ operate until the end of their useful life (i.e., from $t$ to $t + UL/\delta$, where $UL$ corresponds to the operating lifetime of the technology expressed in years and $\delta$ represents the length of one time period, i.e., five years). By running RAPID for different starting years for CDR, we quantify the impact of delaying its deployment to various extents. Delaying actions in time results in more constrained optimization problems, i.e., tighter feasible regions, as the capacities of BECCS and DACCS up to the investigated period are fixed to zero. Consequently, the optimal solution worsens as we solve RAPID for later starting years, as the level of flexibility in the optimization diminishes accordingly (e.g., loss of biomass potential due to the underuse of land). RAPID is deterministic, as it assumes that all model parameters are perfectly known in advance. Nevertheless, we analyzed the effects of key uncertainties on our results by defining three scenarios considering nominal, optimistic, and pessimistic estimates (Uncertainty analysis section in Methods).

**$CO_2$ emissions balance**. A life-cycle thinking approach was followed to estimate all the emissions throughout the life cycle of all the activities involved in the integrated system (power mix and CDR options). We applied life cycle assessment (LCA) principles aligned with the ISO 14040 and ISO 14044 standards[75–77]. We adopted a cradle-to-gate scope where all the emissions data ($E_{ijt}$ in Eqs. 2 and 3) for both the foreground system (power mix and CDR facilities) and the background system (surrounding activities linked to the foreground system) were retrieved from Ecoinvent v3.5[78] (except for the crops cultivation phase, for which we used data from the FEAT database[79]). The Ecoinvent database distinguishes between biogenic and fossil $CO_2$, both included in the life cycle inventory (LCI). The biogenic carbon uptake and the biogenic carbon releases are often unbalanced at the level of activities due to some allocation choices. Our integrated system consumes biomass resources as the main feedstock for the BECCS and biomass power plants; consequently, we need to manually adjust the $CO_2$ balance to quantify precisely the $CO_2$ removed from the atmosphere, whose storage is ensured in the long term. This adjustment applies as well to the DACCS, which also captures atmospheric $CO_2$.

Bearing the above in mind, the biogenic carbon and the $CO_2$ captured with DACCS were tracked manually to adjust the $CO_2$ balance. Hence, we first excluded all the biogenic $CO_2$ from the LCI of all the supply chain activities, thereby considering only the non-biogenic emissions to air (list of elementary flows in Supplementary Information section 2.2.4). Note that omitting the biogenic carbon is a common practice in most LCIA methods focused on climate change under the assumption that the $CO_2$ uptake by biomass via photosynthesis will be eventually released again into the air. Second, the $CO_2$ uptake from the atmosphere via photosynthesis or chemical reactions is modeled as a negative flow of $CO_2$ entering the system. For the biomass resources (i.e., energy crops and residues from agriculture and forestry activities), the $CO_2$ uptake is estimated from the carbon and water content (Supplementary Table 25). Finally, these negative $CO_2$ flows are tracked along the supply chains by accounting for the flows leaving the system as positive flows (e.g., biomass losses, uncaptured $CO_2$, or other leakages), to precisely establish the $CO_2$ emissions balance. More details on the emissions data and associated references are presented in the Supplementary Information in section 2.2.4.

**Technology deployment**. Diffusion rate constraints preclude solutions that are not consistent with the deployment rates often found in similar technologies. BECCS is constrained by both the availability of biomass resources and the maximum deployment rate (as well as by the storage capacity). DACCS, however, is only limited by the maximum diffusion rate (and the storage capacity). All power technologies are constrained by the resource availability and diffusion rate, while the model limits, in turn, the penetration of intermittent renewables to ensure the grid's reliability. In essence, the diffusion constraints (Supplementary Equations 24–29) impose limits to the rate at which technologies can scale up under the assumption that external factors constrain the speed of deployment, e.g., market forces, competition issues, infrastructure adaptation, learning rates, or social acceptance[51,69]. The diffusion constraint, therefore, relates the capacity in period $t$ to the capacity already installed in the previous period $t$-1, i.e., $cap_t^{Avail} \leq cap_{t-1}^{Avail}(1 + \gamma)^{\delta}$ where $\gamma$ is the maximum annual growth rate ($\gamma \in [0, 1]$) and $\delta$ is the duration of the period. In RAPID, we assumed an annual growth rate of 20% ($\gamma$ equals 0.2 and $\delta$ equals 5 years) based on the historical diffusion rates of energy-related technologies[69]. According to the diffusion constraint, technologies with larger installed capacities can diffuse faster. The maximum deployment rates of technologies over time and their initial capacity are shown in Supplementary Fig. 4.

**Uncertainty analysis**. We carry out an uncertainty analysis to quantify how uncertainties in the economic and emission parameters affect the outcome of RAPID. In particular, we perform an *a posteriori* analysis of these uncertainties to establish confidence intervals for the costs and $CO_2$ emissions in each solution. Hence, besides the base case scenario considering nominal values, we define best-case and worst-case scenarios assuming optimistic and pessimistic values for the parameters, respectively.

Uncertainties in the inventory data are modeled based on Ecoinvent[80]. We used Simapro v9.0[81] to generate 1,000 samples from the probability functions of the uncertain emissions via Monte Carlo sampling; from these samples, we defined the optimistic and pessimistic scenarios, which consider the mean value of the emissions parameters minus and plus two times the standard deviation, respectively (Supplementary Table 27). For the cost uncertainty, we defined the optimistic and pessimistic scenarios considering low and high estimates, respectively, for the CAPEX and OPEX parameters sourced from the literature (Supplementary Tables 10 and 11). The uncertainty analysis results for the costs are provided in Supplementary Tables 43-50, while for the emissions, the results are depicted in Fig. 1b with a shaded area covering the best- and worst-case scenarios.

We also performed a sensitivity analysis to study the effects of uncertainties affecting the availabilities of marginal land and biomass residues on the results. We defined various scenarios with increased and reduced potentials considering different percentages over the central estimates shown in Supplementary Tables S37 and S38 for marginal land and biomass residues (i.e., ±50, ±25%, ±10%). The RAPID model was then solved for the said scenarios to evaluate the impact of these uncertainties on the outcome of the optimization. The results of the sensitivity analysis on the biomass potentials are shown in Supplementary Fig. 2.

## Data availability

The data supporting the findings of this study are available within the supplementary information document and supplementary data files.

## Code availability

The computer code supporting our analysis is available from the corresponding author upon request.

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

## Acknowledgements

Á.G-M thanks the Spanish Ministry of Science, Innovation, and Universities for the financial support through the Beatriz Galindo Program (BG20/00074). J.A.C acknowledges financial support from the Generalitat Valenciana under project PRO-METEO 064/2020.

## Author contributions

Á.G.-M. conceived the research. D.V., Á.G.-M. and G.G.-G. designed the study. D.V. and Á.G.-M. developed the model formulation, carried out the analyses, and created the illustrations. Á.G.-M. and G.G.-G. wrote the paper. Á.G.-M., D.V., S.C., N.M.D., J.A.C. and G.G.-G. contributed to identifying data, interpreting the results, and revising.

## Competing interests

The authors declare no competing interests.
