## [Peer Review File · Nature Communications]

Reviewer comments, initial round review -

Reviewer #1 (Remarks to the Author):

1. This manuscript addresses extremely important questions in terms of the potential timing of deployment of carbon dioxide removal options. Most members of the climate community now recognize the necessity for large-scale deployment of CDR/NETs options, but the implications in terms of cost increases with delay has only been briefly discussed in other studies, and the question of whether the potential TOTAL capacity might be diminished by delay is a novel one, making this contribution very valuable. The questions of economics and capacity when mapped against the issue of temporal deployment has very substantial ramifications for policymaking in this context, including R&D and incentives for deployment;
2. The moral hazard argument at the outset is important in terms of how, or if, CDR will receive public and political support, but it strikes me as constructed in a confused fashion here. I'm not sure that the uncertainties about the potential viability of CDR totally explains the moral hazard concerns of some. For example, many in the NGO community also express concern that even IF CDR options can be deployed at large-scale, we still need, as you argue, full-throated decarbonization. However, many fear that the prospects of CDR could detract from keeping our pedal to the metal in reducing emissions, which is a different argument that the trepidation about what happens if CDR is never deployed at scale;
3. I wonder if the scenarios in this study outline the full range of potential options, most notably the role of afforestation and reforestation, perhaps coupled with biochar. Would this change the calculus in terms of potential "lost" biomass? While I agree that some other options, e.g. enhanced mineral weathering, and marine-based options remain speculative, AF/RF was front and center in the IPCCAR5, and will play a major role in AR6 assumptions of CDR. Might this chance your scenarios substantially? You acknowledge on p. 5 that these options might INCREASE the CDR potential, but they might also compete with BECCS, for example, for resources, but perhaps in a way that is optimal in terms of total sequestration and co-benefits;
4. The question of the "loss" of biomass residues is a bit of a tricky one if one doesn't factor in the potential of biochar, which again, might be integrated with BECCS, but is not considered in this study;
5. I think the question of the diffusion factor of DAC is a very interesting one. Obviously, there are some serious points of contention about potential rates of diffusion, but it's an important point to make in terms of the implications of delay that hasn't been discussed much in the past;
6. I would have liked a bit more discussion of the "under use" of "marginal" lands in terms of BECCS given the fact that many studies conclude that much of the purportedly "abandoned" or "marginal" land is actually being used in livelihoods. How well is your model capturing sustainability considerations?
7. Again, in terms of sustainability, is the use of large swathes of agricultural land in Europe for miscanthus product likely to result in substantial rises in prices of agricultural foodstocks for the most vulnerable? Again, there's a reference to "marginal" land, but I both question how much of it actually is marginal, and how much "marginal" land might have to be used for agriculture in the future as population increases increase the demand for food (perhaps not so much in the EU, but elsewhere where large-scale BECCS is being discussed);
8. I liked the granular analysis of which countries in Europe would most likely deploy BECCS and DAC, respectively, though I wonder if biomass export is likely to flow predominantly between European countries. Suggestions that much of the biomass might come from the African continent might pose sustainability and/or social justice questions;
9. The question of the EU's alleged "fair responsibility" in terms of deploying CDR remains a contested concept, i.e. does it flow from principles of common but differentiated responsibility under the UNFCCC or Paris, does it actually exist at all? However, it's an important additional potential rationale for early deployment; thus, I would move this discussion into the introduction, rather than in the Conclusion, where it appears more of an afterthought. In this context, you might look at the recent comments by Indian officials, arguing that developed states should shoulder this responsibility;
10. Another afterthought in the Conclusion was the question of whether the potential for CCS to crowd out BECCS/DACs in terms of storage capacity, providing an additional rationale for phasing

out fossil fuels quickly is an interesting one.

Reviewer #2 (Remarks to the Author):

This manuscript explores the consequences of delayed CDR deployment in the European power sector, which are namely significant economic losses and depreciated cumulated carbon dioxide removal potentials putting climate targets at risk. The authors go an important step beyond previous assessments, which mainly explored delayed mitigation effort but, to my knowledge, never explicitly explored the delay of CDR deployment therein. The paper is concise and well written, the scenario set-up is good and well described with clear figures, and its novelty and timely contribution merit publication in Nature Communications.

However, there are some points I would like to raise:

General remark

The authors investigate the feasibility of indicative CDR targets to be reached by the end of the century, while the European Green Deal is setting targets of net-GHG neutrality for the EU in 2050. When the methods described in this article could be used to explore the feasibility of reaching GHG neutrality (or as a proxy, indicative net negative emissions from the power sector) in 2050 with delayed CDR action, it would greatly improve the political relevance and be extremely valuable for the current debate.

Regional implications for the energy systems

Fig3b) Biomass is extremely valuable in decarbonisation pathways but a scarce resource. There are many studies that allocate biomass use outside of the electricity generation. See Bauer 2018 for example. Could you perform a sensitivity analysis with reduced biomass availability to better reflect sectoral competition for biomass utilisation and concerns on sustainable biomass availability?

Minor remarks:

Introduction

74| I would recommend to add "to achieve a net CO2 emissions target in the power sector" as other sector's emissions are not fully covered in the model used in this article.

Results and discussion

Fig 1b) just for clarification: Why is BECCS deployment not restricted by diffusion rates, while DACCS deployment is critically limited in the delayed cases?

152-154| DACCS diffusion rate and initial capacity are given. Could you add your BECCS initial capacity assumptions as well?

Regional implications for the energy systems

Fig 3a) Having such a large share of nuclear in the German power mix is extremely unlikely, but probably necessary in the RAPID model in the absence of storage technologies. I don't expect the fundamental results concerning CDR deployment to change much with different power system technologies available, but it would be good to mention alternatives presented in the literature. (e.g. Victoria et al. 2020, Gerbaulet et al. 2020, Plessmann et al. 2017, Pietzcker et al. 2021)

Supplemental Material

132 | "back-up" typo

466 | Table 10, Geothermal 2020(p1) typo

821 – 824 | Table 47 and Table 48 are identical

Supplementary Fig. 2 | It would be nice to have a dotted version of BECCS using the Maximum achievable carbon removal plotted to the sequestration axis together with DACCS.

Bauer, Nico, et al. "Global energy sector emission reductions and bioenergy use: overview of the bioenergy demand phase of the EMF-33 model comparison." *Climatic Change* (2018): 1-16.
Victoria, M., Zhu, K., Brown, T. et al. Early decarbonisation of the European energy system pays off. *Nat Commun* 11, 6223 (2020). <https://doi.org/10.1038/s41467-020-20015-4>
Gerbaulet, C., von Hirschhausen, C., Kemfert, C., Lorenz, C., & Oei, P. Y. (2019). European electricity sector decarbonization under different levels of foresight. *Renewable energy*, 141, 973-987.

Plessmann, G., & Blechinger, P. (2017). How to meet EU GHG emission reduction targets? A model based decarbonization pathway for Europe's electricity supply system until 2050. *Energy Strategy Reviews*, 15, 19-32.

Pietzcker, R. C., Osorio, S., & Rodrigues, R. (2021). Tightening EU ETS targets in line with the European Green Deal: Impacts on the decarbonization of the EU power sector. *Applied Energy*, 293, 116914.

Referees' comments in *blue* - Replies in black - Actions in **bold**.
Indicated page and line refer to the revised manuscript and ESI.

Response to Comments from Reviewer #1

Overall Comment: *1. This manuscript addresses extremely important questions in terms of the potential timing of deployment of carbon dioxide removal options. Most members of the climate community now recognize the necessity for large-scale deployment of CDR/NETs options, but the implications in terms of cost increases with delay has only been briefly discussed in other studies, and the question of whether the potential TOTAL capacity might be diminished by delay is a novel one, making this contribution very valuable. The questions of economics and capacity when mapped against the issue of temporal deployment has very substantial ramifications for policymaking in this context, including R&D and incentives for deployment;*

Authors: We greatly appreciate that the Reviewer considers our manuscript relevant and timely and that she/he found our contribution novel and valuable. We sincerely appreciate her/his thoughtful comments and suggestions that helped improve the manuscript.

Comment 1: *2. The moral hazard argument at the outset is important in terms of how, or if, CDR will receive public and political support, but it strikes me as constructed in a confused fashion here. I'm not sure that the uncertainties about the potential viability of CDR totally explains the moral hazard concerns of some. For example, many in the NGO community also express concern that even IF CDR options can be deployed at large-scale, we still need, as you argue, full-throated decarbonization. However, many fear that the prospects of CDR could detract from keeping our pedal to the metal in reducing emissions, which is a different argument that the trepidation about what happens if CDR is never deployed at scale;*

Authors: We thank the Reviewer for raising this point. Indeed, we fully agree that the way in which it was presented was not fully clear. **We have rewritten the sentence to clarify the message we wanted to convey (see page 2, lines 53-57).**

Comment 2: *3. I wonder if the scenarios in this study outline the full range of potential options, most notably the role of afforestation and reforestation, perhaps coupled with biochar. Would this change the calculus in terms of potential "lost" biomass? While I agree that some other options, e.g. enhanced mineral weathering, and marine-based options remain speculative, AF/RF was front and center in the IPCCAR5, and will play a major role in AR6 assumptions of CDR. Might this chance your scenarios substantially? You acknowledge on p. 5 that these options might INCREASE the CDR potential, but they might also compete with BECCS, for example, for resources, but perhaps in a way that is optimal in terms of total sequestration and co-benefits;*

Authors: Thanks for raising this comment. We agree with the Reviewer on the importance of considering a comprehensive portfolio of CDR alternatives, yet we focused on BECCS and DACS for the reasons below. We also provide next an in-depth analysis of the potential implications of extending the study to embrace other CDR alternatives.

Weak links to the power mix:

- The analysis could include afforestation /reforestation (AF/RE), biochar, and even BECCS for fuel transportation (e.g., bioethanol with CCS or biomass gasification with CCS to hydrogen). However, we focused on BECCS (to power) and DACCS because they display strong links with the power system. Consequently, integrating them at the earliest is key to designing future power systems optimally. Thus, BECCS and DACCS technologies undoubtedly lie within the scope of the RAPID model developed *ad hoc* to address the questions motivating this work. In contrast, AF and RE are natural-climate CDR solutions displaying much weaker links to the power sector.
- The same applies to biochar, lying between natural and technological approaches, due to its main use as a soil amendment. Biochar is usually generated in a slow pyrolysis process that maximizes biochar production and minimizes bio-oil pyrolytic gas. The pyrolysis process itself consumes a small amount of electricity, so its link with the power system is also weak.

Uncertainties surrounding the estimation of the sequestration potential and low technology readiness level:

BECCS, DACCS and AF/RE are already included in most Integrated Assessment Models (in the IPCC Special Report on Global Warming of 1.5°C, i.e., IPCC SR1.5). However, at present, only BECCS and DACCS are being pursued at large scale. Furthermore, both are coupled with geological storage (e.g., in depleted oil and gas reservoirs, coal beds, and saline aquifers) to ensure the CO₂ sequestration in the long-term with low vulnerability. In contrast, AF/RE (and, to a lesser extent, biochar) suffer from permanence issues and other shortcomings such as the saturation of sinks, vulnerability to disturbances, and accounting difficulties (Fuss et al. (2018)).

Notably, forestation provides carbon sequestration susceptible to many factors such as wildfires, drought, or diseases that risk the long-term CO₂ storage (Anderegg et al.(2020)). For example, the massive forest fires in Australia or California have released vast amounts of carbon in recent years. Moreover, AF and RE suffer from accounting issues due to the difficulties in establishing the crediting lifetime precisely. Since the plantation project, it takes years (e.g., 40 years) until the forest is mature and the sink saturates, requiring sustainable forest management long after forestation occurs. Hence, a fair comparison of the CO₂ removal of BECCS, AR/RE requires accounting for the sustainable management of a forest for a very long period (i.e., in the order of thousands of years) to match the long-term geological storage. Notably, the credits of AR/RE projects may accrue over more extended periods relative to those for BECCS and DACCS projects (i.e., much longer than the time horizon considered in our study). This mismatch in temporal scales poses a significant challenge when combining all these technologies in a single model.

- Likewise, the carbon sequestration capacity of biochar is highly variable; its long-term stability depends on the feedstock quality, the pyrolysis process conditions, and the soil and climate conditions (Gurwick et al., 2013). Hence, estimating the potential for biochar is challenging because i) the residence time in the field is highly uncertain (Wang et al. (2016)), and ii) the application rate depends on the assimilation capacity of the soil and the sink saturation, among other factors.

Moreover, biochar has not yet been applied on a large enough scale (low maturity) but rather at the laboratory scale mostly. Therefore, there are large uncertainties regarding

costs, potentials, long-term removal efficacy, and potential side effects. For example, land management and carbon accounting would be required long after the biochar application to soil occurs to ensure a fair comparison with other CDR options. Similarly, estimating the “true” costs of biochar is very challenging due to its side benefits, such as positive effects on soil quality, nutrients, and water cycles.

Lower sequestration potential compared to BECCS:

- AR refers to the growth of forest in non-forest biomes, which would compete for marginal land with the cultivation of energy crops for BECCS (Griscom et al. (2017)). It has been shown that growing dedicated energy crops in the marginal land available provides greater CO₂ removal capacity per hectare than growing new forests. According to Smith et al. 2016, the removal efficiency with *Miscanthus* ranges from 5.83 to 8.59 t Ceq./ha/yr. In contrast, the mean accrual rate for afforestation/reforestation is around 3.4 tCeq./ha/yr. Therefore, we did not consider afforesting marginal land because bioenergy crops yield higher sequestration rates per hectare.
- Biochar production has potential for diverse applications beyond soil amendment, such as generating power by combusting the co-products bio-oil and pyrolytic gas. Hence, to further support our decision to exclude biochar from the study, we have investigated two scenarios for biochar production delivering both biochar and electricity. We have compared their net life-cycle CO₂ emissions with BECCS to power, considering the biomass resources studied in our work and a 100-year time horizon. The data for the pyrolysis process were adapted from Peters et al. (2015). In scenario *BC subst. sand*, biochar substitutes sand in building materials (Werner et al., 2018), and in scenario *BC to soil*, the biochar is applied to the soil. The latter considers a biochar lifetime of 1000 years (Schmidt et al., 2019), which releases to air 10% of the sequestered carbon as CO₂ within a 100-year time horizon (assuming a constant carbon degradation rate). When biochar is used as a sand replacement, it remains sequestered within the building material during the considered timeframe. Our results, shown below in Figure R1, demonstrate that BECCS is a more efficient CDR strategy than biochar since it provides a higher CDR potential per tonne of biomass. Our findings are aligned with previous works, such as that by Patrizio et al.(2021). Note, however, that our comparison only considers removal potential while other criteria should also be accounted for. Moreover, the CO₂ balance might be highly case- and site-specific.

No competition for resources with BECCS and DACCS:

- RE refers to restocking existing forests; therefore, it will not compete with the marginal land reserved for BECCS. In our recent work (Pozo et al. (2020)), we estimated the CDR potentials considering reforestation and natural forest management practices for the EU countries based on Griscom et al. (2017). These CDR options include converting non-forest areas (with less than 25% of tree cover) to forest and implementing improved practices in forests under timber production, such as delaying harvests or adopting reduced-impact logging practices. Our RE estimates could be added to the countries' CDR potentials, increasing the amount of CDR delivered (but subject to the uncertainties discussed above, i.e., risks, permanence, saturation, and accounting issues). This potential amounts to 0.91 Gt CO₂ per year until the sink saturates (i.e., 30 years when the forest reaches the steady-state), while improved management practices such as delaying harvests or adopting reduced-impact logging would further remove 0.07 Gt CO₂ by 2100 (Pozo et al. (2020)). Adding them would shift the CDR potential curve in Fig. 1 of the main manuscript downwards. However, note that the optimal

power system would be the same because these practices do not compete for resources with DACCS and BECCS (neither for the land nor for geological storage). Therefore our main conclusions and findings would remain the same, particularly those related to the power system. This answers the second question posed by the Reviewer in her/his comment since considering RE would add CDR potential, but the optimal power system and the implications of delayed actions would remain the same.

Regarding the Reviewer's first question, including AF/RE and biochar would not prevent the biomass losses due to inaction. CDR options are deployed from a particular year within the time horizon. Reforestation would not compete for any resource but rather provide additional natural carbon sinks. In contrast, biochar would compete with BECCS for the biomass resources available. However, as it happens with BECCS, not producing biochar during inactive periods would also result in the loss of biomass potential due to these idle times.

We also note that when biochar or other negative emission technology or practice becomes more mature and the uncertainties surrounding it diminish, it could be added to the model based on the new knowledge acquired.

We have clarified further this modeling assumption in the text (page 18, lines 485-486). We have also highlighted that considering these CDR options would provide greater flexibility to exploit regional endowments and deliver CDR (see page 6, lines 149-156). We have also included estimates for reforestation and natural forest management practices (see page 6, lines 152-156).

Finally, we hope that the Reviewer finds our response and changes appropriate to address her/his concern. We prefer to keep the original scope of our model as it is, focusing on the CDR options with strong links to the power sector, key for achieving carbon neutrality. In any case, we will be happy to implement further changes in the model if our response does not fully address the Reviewer's concerns. However, these would likely result in more pronounced uncertainties.

Figure R1. Total CO₂ emissions of combustion-BECCS and two pyrolysis scenarios based on the application of biochar to the soil (BC to soil) and the substitution of sand with biochar in building materials (BC subst. sand), for the considered biomass resources: **a** forestry residues, **b** agricultural prunings, **c** straw residues, **d** willow, **e** switchgrass, **f** miscanthus.

Fuss, S., Lamb, W. F., Callaghan, M. W., Hilaire, J., Creutzig, F., Amann, T., ... & Minx, J. C. (2018). Negative emissions—Part 2: Costs, potentials and side effects. *Environmental Research Letters*, 13(6), 063002.

Anderegg, W. R., Trugman, A. T., Badgley, G., Anderson, C. M., Bartuska, A., Ciais, P., ... & Randerson, J. T. (2020). Climate-driven risks to the climate mitigation potential of forests. *Science*, 368(6497).

Griscom, B. W., Adams, J., Ellis, P. W., Houghton, R. A., Lomax, G., Miteva, D. A., ... & Fargione, J. (2017). Natural climate solutions. *Proceedings of the National Academy of Sciences*, 114(44), 11645-11650.

Gurwick, N. P., Moore, L. A., Kelly, C. & Elias, P. (2013). A Systematic Review of Biochar Research, with a Focus on Its Stability in situ and Its Promise as a Climate Mitigation Strategy. *PLoS One* 8.

Wang, J., Xiong, Z., & Kuzyakov, Y. (2016). Biochar stability in soil: meta-analysis of decomposition and priming effects. *Gcb Bioenergy*, 8(3), 512-523.

Pozo, C., Galán-Martín, Á., Reiner, D. M., Mac Dowell, N., & Guillén-Gosálbez, G. (2020). Equity in allocating carbon dioxide removal quotas. *Nature Climate Change*, 10(7), 640-646.

Peters, J. F., Iribarren, D., & Dufour, J. (2015). Biomass pyrolysis for biochar or energy applications? A life cycle assessment. *Environmental science & technology*, 49(8), 5195-5202.

Patrizio, P., Fajardy, M., Bui, M., & Mac Dowell, N. (2021). CO₂ mitigation or removal, the optimal uses of biomass in energy systems decarbonization. *iScience*, 102765.

Schmidt, H. P. et al. (2019). Pyrogenic carbon capture and storage. *GCB Bioenergy* 11, 573–591.

Smith, P., Davis, S. J., Creutzig, F., Fuss, S., Minx, J., Gabrielle, B., ... & Yongsung, C. (2016). Biophysical and economic limits to negative CO₂ emissions. *Nature climate change*, 6(1), 42-50.

Werner, C., Schmidt, H. P., Gerten, D., Lucht, W. & Kammann, C. (2018). Biogeochemical potential of biomass pyrolysis systems for limiting global warming to 1.5 °C. *Environ. Res. Lett.* 13.

Comment 3: 4. *The question of the "loss" of biomass residues is a bit of a tricky one if one doesn't factor in the potential of biochar, which again, might be integrated with BECCS, but is not considered in this study;*

Authors: Thanks for this comment. We refer the Reviewer to our response to the previous comment justifying why we excluded biochar from our analysis. Note that, in our modeling framework, we assume that CDR options cannot be deployed during periods of inaction. Hence, even if we included biochar, this technology would not be deployed during inactive periods. Therefore, even considering biochar, biomass residues would not be mobilized during inactive periods, and the marginal land available to grow dedicated crops would not be exploited, leading to a potential loss. **We have clarified our modeling approach in the Methods section (see page 18, lines 485-486).**

Comment 4: 5. *I think the question of the diffusion factor of DAC is a very interesting one. Obviously, there are some serious points of contention about potential rates of diffusion, but it's an important point to make in terms of the implications of delay that hasn't been discussed much in the past;*

Authors: We thank the Reviewer for this positive comment. Indeed, the diffusion rate of technologies is very seldom considered in the literature, despite critically constraining the capacity expansions of technologies. This simplification can lead to unrealistic scenarios showing deployment rates never found in practice.

Comment 5: 6. *I would have liked a bit more discussion of the "under use" of "marginal" lands in terms of BECCS given the fact that many studies conclude that much of the purportedly*

"abandoned" or "marginal" land is actually being used in livelihoods. How well is your model capturing sustainability considerations?

Authors: We greatly appreciate the comment made by the Reviewer, which we have addressed through further clarifications and a sensitivity analysis, as discussed next. In our model, based on sustainability criteria, we consider marginal land available for dedicated energy crops, which avoids conflicts with food and biofuels while being respectful with the environment. Notably, to derive our estimates for marginal land available, we followed a conservative approach previously employed in a recent article (Pozo et al. (2020)). We rely on the GIS results provided by Cai et al. (2010) (considering the most conservative scenario), which we downgraded by 69% according to Fritz, et al. (2013), which further reduced the original estimates for marginal land.

Nevertheless, the land available remains uncertain and might be lower due to other competitive uses, as pointed out by the Reviewer. However, other authors argue that more land might eventually become available due to improvements in agriculture or dietary changes (Röös et al. (2017), Griscom et al. (2017)). Hence, we have compared our estimates with the results derived from a recent H2020 project (Marginal lands for Growing Industrial Crops: Turning a burden into an opportunity [1]). We found that their mapped marginal land area is much larger than our estimates, confirming that we employed conservative estimates (e.g., their marginal land area in Poland is 25,796 km² while our estimates are 7,371 km², i.e., our estimates are around 30% lower). **In the new revised version, we have better explained our calculations for marginal land in the Supplementary Information (pages S40 and S41, lines 739-758) while adding a brief discussion in the main text (pages 5 and 6, lines 129-144).**

Our calculations are based on sustainable practices (e.g., soil conservation and biodiversity protection) regarding our estimates for the agricultural and forestry residues. We also subtract other competitive uses of such residues, such as using straw for animal bedding. **We have clarified that we considered sustainable potentials for biomass residues in the Supplementary Information (page S41, lines 764-768).**

Moreover, acknowledging that the marginal land and biomass residues availability is highly uncertain, we have performed a sensitivity analysis on these parameters. Details on the sensitivity analysis can be found in the Methods section (Uncertainty analysis, page 20, lines 557-562), while the results are presented in the Supplementary Information document (Supplementary Fig. 2). We now briefly discuss these additional results in the main text (see page 6, lines 138-145). We note that the main conclusions of the work would remain the same, even considering these uncertainties.

Pozo, C., Galán-Martín, Á., Reiner, D. M., Mac Dowell, N., & Guillén-Gosálbez, G. (2020). Equity in allocating carbon dioxide removal quotas. Nature Climate Change, 10(7), 640-646.

Cai, X., Zhang, X. & Wang, D. Land availability for biofuel production. Environ. Sci. Technol. 45, 334–339 (2010).

Fritz, S. et al. Downgrading recent estimates of land available for biofuel production. Environ. Sci. Technol. 47, 1688–1694 (2013).

Röös, E. et al. Greedy or needy? Land use and climate impacts of food in 2050 under different livestock futures. Glob. Environ. Change 47, 1–12 (2017).

Griscom, B. W., Adams, J., Ellis, P. W., Houghton, R. A., Lomax, G., Miteva, D. A., ... & Fargione, J. (2017). Natural climate solutions. *Proceedings of the National Academy of Sciences*, 114(44), 11645-11650.

[1] Marginal lands for Growing Industrial Crops: Turning a burden into an opportunity. <https://cordis.europa.eu/project/id/727698/results/es>

<https://www.arcgis.com/apps/webappviewer/index.html?id=a813940c9ac14c298238c1742dd9dd3c>

Comment 6: *7. Again, in terms of sustainability, is the use of large swathes of agricultural land in Europe for miscanthus product likely to result in substantial rises in prices of agricultural foodstocks for the most vulnerable? Again, there's a reference to "marginal" land, but I both question how much of it actually is marginal, and how much "marginal" land might have to be used for agriculture in the future as population increases increase the demand for food (perhaps not so much in the EU, but elsewhere where large-scale BECCS is being discussed);*

Authors: We thank the Reviewer for raising this point, addressed via further clarifications and a sensitivity analysis. As stated in our previous response to Comment 6, we employed conservative estimates for marginal land availability. The current and future marginal land available is highly uncertain, and its use remains unclear. Notably, some authors claim that the current cropland will suffice to feed the future population due to agricultural intensification or dietary changes (Röös et al. (2017)) without the need to rely on marginal land.

We followed the same procedure for our marginal land calculations as in our previous work (Pozo et al. (2020)). We first calculated the marginal land available at the EU country level following a GIS approach considering the results from Cai et al. (2010). These authors estimated the marginal land considering soil productivity, slope, climate, and land cover criteria. In particular, we considered the most conservative scenario (scenario S1) from this original work, which includes at least part of the abandoned, wasted, or idle agricultural land and some small crop fields. Then, we adjusted the marginal land availability by downgrading the original estimates by 69%, according to Fritz, S. et al. (2013). **We have clarified this procedure in the Supplementary Information (pages S40 and S41, lines 739-758). We have also performed a sensitivity analysis on the marginal available and biomass residues. The details on the sensitivity analysis are presented on page 20, lines 557-562, and the results are shown in the new Supplementary Fig. 2 (page S49), accompanied by a brief discussion in the main text (page 6, lines 138-145).**

We argue that considering marginal land addresses the issue of land competition for food production of BECCS, with several authors concluding that deploying BECCS sustainably is feasible (Rogelj et al. (2016), Creutzig et al. (2015)). However, land-use competition may still arise because edible crops might be cultivated in degraded land (despite being economically unappealing), so deploying BECCS at scale will always be controversial. Nevertheless, we think that our conservative estimates might (arguably) represent marginal land “unsuitable” for food production, thereby minimizing land-use competition and other adverse side effects such as biodiversity loss. **Hence, besides performing a new sensitivity analysis on the marginal land available, we have introduced a brief discussion in this regard while referencing the works by Rogelj et al. (2016) and Creutzig et al. (2015) (see pages 5 and 6, lines 130-145)**

Pozo, C., Galán-Martín, Á., Reiner, D. M., Mac Dowell, N., & Guillén-Gosálbez, G. (2020). Equity in allocating carbon dioxide removal quotas. *Nature Climate Change*, 10(7), 640-646.

Cai, X., Zhang, X. & Wang, D. Land availability for biofuel production. *Environ. Sci. Technol.* 45, 334–339 (2010).

Fritz, S. et al. Downgrading recent estimates of land available for biofuel production. *Environ. Sci. Technol.* 47, 1688–1694 (2013).

Rogelj, J., Den Elzen, M., Höhne, N., Fransen, T., Fekete, H., Winkler, H., ... & Meinshausen, M. (2016). Paris Agreement climate proposals need a boost to keep warming well below 2 C. *Nature*, 534(7609), 631-639.

Creutzig, F. et al. Bioenergy and climate change mitigation: an assessment. *GCB Bioenergy* 7, 916–944 (2015)

Comment 7: 8. *I liked the granular analysis of which countries in Europe would most likely deploy BECCS and DAC, respectively, though I wonder if biomass export is likely to flow predominantly between European countries. Suggestions that much of the biomass might come from the African continent might pose sustainability and/or social justice questions;*

Authors: We thank the Reviewer for this valuable comment. In our work, we explicitly mention and highlight that we only consider EU domestic potentials for biomass resources (i.e., residues and marginal land) and onshore geological sites. Under this assumption and considering a full cooperative strategy, biomass trading between EU countries would likely emerge to exploit biomass resources further and reduce the reliance on electricity imports (see Fig. 4a and page 13, lines 345-348). Nevertheless, as pointed out by the Reviewer, imported biomass might be preferred over indigenous resources when better crop yields, carbon intensities of power mixes, and climate data offset the advantages of the lower within-EU transport distances (Fajardy et al. (2018)). **Hence, following the Reviewer's comment, we have added in the text that importing biomass from abroad or exporting CO₂ could pose sustainability and social justice questions besides governance issues (see page 6, line 146-149).**

Fajardy, M., Chiquier, S., & Mac Dowell, N. (2018). Investigating the BECCS resource nexus: delivering sustainable negative emissions. *Energy & Environmental Science*, 11(12), 3408-3430.

Comment 8: 9. *The question of the EU's alleged "fair responsibility" in terms of deploying CDR remains a contested concept, i.e. does it flow from principles of common but differentiated responsibility under the UNFCCC or Paris, does it actually exist at all? However, it's an important additional potential rationale for early deployment; thus, I would move this discussion into the introduction, rather than in the Conclusion, where it appears more of an afterthought. In this context, you might look at the recent comments by Indian officials, arguing that developed states should shoulder this responsibility;*

Authors: We thank the Reviewer for this comment. Indeed, in our recent article (Pozo et al. (2020)), we allocated CDR quotas at the country level according to Responsibility, Capability, and Equality principles, all consistent with the UNFCCC's call. We agree that ensuring fair

contributions could also be a potential rationale to expedite CDR action and, accordingly, **we have now highlighted this point in the introduction (page 2, lines 64-65).**

Pozo, C., Galán-Martín, Á., Reiner, D. M., Mac Dowell, N., & Guillén-Gosálbez, G. (2020). Equity in allocating carbon dioxide removal quotas. Nature Climate Change, 10(7), 640-646.

Comment 9: *10. Another afterthought in the Conclusion was the question of whether the potential for CCS to crowd out BECCS/DACs in terms of storage capacity, providing an additional rationale for phasing out fossil fuels quickly is an interesting one.*

Authors: Thanks for the comment. We do think that this is an important take-home message that was never discussed before in the literature to our best knowledge. Our results show that inaction on CDR and late phasing out of fossil fuels with facilities being retrofitted with CCS may exhaust the geological sites. Competition issues might arise at the regional level because suitable geological sites might be depleted with fossil CO₂ captured in existing plants rather than carbon from the air. For example, the US tax credit known as 45Q is currently incentivizing the use of captured CO₂ for enhanced oil recovery (EOR) (\$50/t of CO₂ permanently sequestered and \$35/t CO₂ captured and used for enhanced oil recovery). Capturing biogenic CO₂ at bioethanol plants, for example, net removes carbon from the atmosphere. However, fossil-based coal and gas-fired power plants or the cement industry could also take advantage of the credit, avoiding emissions of fossil CO₂ but failing to remove carbon from the air.

Response to Comments from Reviewer #2

Overall Comment: *This manuscript explores the consequences of delayed CDR deployment in the European power sector, which are namely significant economic losses and depreciated cumulated carbon dioxide removal potentials putting climate targets at risk. The authors go an important step beyond previous assessments, which mainly explored delayed mitigation effort but, to my knowledge, never explicitly explored the delay of CDR deployment therein. The paper is concise and well written, the scenario set-up is good and well described with clear figures, and its novelty and timely contribution merit publication in Nature Communications.*

However, there are some points I would like to raise:

Authors: We greatly appreciate that the Reviewer considers our manuscript interesting, novel, and well-presented. We are glad that the Reviewer finds our work suitable for publication in *Nature Communications*. We thank the Reviewer for her/his constructive comments, which helped us to improve our work.

Comment 1: *General remark*

The authors investigate the feasibility of indicative CDR targets to be reached by the end of the century, while the European Green Deal is setting targets of net-GHG neutrality for the EU in 2050. When the methods described in this article could be used to explore the feasibility of reaching GHG neutrality (or as a proxy, indicative net negative emissions from the power sector) in 2050 with delayed CDR action, it would greatly improve the political relevance and be extremely valuable for the current debate.

Authors: We thank the Reviewer for this insightful and constructive comment. We agree with the Reviewer that the EU climate neutrality goal by 2050 is very relevant for policymaking. Accordingly, to provide further insight into the implications of inaction on CDR for the EU climate neutrality strategy, we have re-run our model for the 2050 horizon. Similar to the 2100 case, we have found that delaying CDR affects the cost and feasibility of reaching the 2050 target sought substantially. Notably, delaying CDR actions beyond 2040 might prevent the EU power sector from reaching carbon neutrality 2050, while inaction would increase the associated costs in the range of 0.05–0.39 trillion EUR per year. **We have provided the new results for the 2050 horizon in the Supplementary Information document (see new Supplementary Fig. 3 in page S50, shown also below for convenience). We have briefly discussed these new results at the end of the first section of the article (page 7, lines 187-192), highlighting their relevance in the EU Green Deal context. Moreover, in the Methods section, we have explicitly mentioned that our RAPID model is flexible, and the horizon could be easily modified at will depending on the needs (page 17, lines 431-432).**

Supplementary Fig. 3. Implications on costs and emissions of delayed-actions on CDR considering different starting points for BECCS and DACCS deployment in the context of the EU climate neutrality goals by 2050 (x-axis). Subplot **a** shows the minimum costs of the EU power system associated with increasing CDR targets. Subplot **b** shows the maximum cumulative net CDR that could be attained deploying BECCS and DACCS from a particular point in time onwards (green profile only with BECCS, blue with DACCS, and yellow considering both BECCS and DACCS). Dots correspond to the optimal solutions for the 5-year time steps starting in 2020 and ending in 2020. The shaded area in subplot **b** indicates the ranges of the results considering uncertainties in the life cycle CO₂ emissions (i.e., $\mu \pm 2\sigma$, Methods for details on the uncertainty analysis). The pie charts illustrate the proportion of gross CDR provided with BECCS and DACCS, respectively.

Comment 2: Regional implications for the energy systems

Fig3b) Biomass is extremely valuable in decarbonisation pathways but a scarce resource. There are many studies that allocate biomass use outside of the electricity generation. See Bauer 2018 for example. Could you perform a sensitivity analysis with reduced biomass availability to better

reflect sectoral competition for biomass utilisation and concerns on sustainable biomass availability?

Authors: We thank the Reviewer for this valuable comment, which we addressed with further clarifications and a sensitivity analysis, as requested. Indeed, several economic sectors will compete for biomass resources due to their inherent versatility (e.g., biofuels from biomass). Moreover, biomass availability will remain uncertain due to land competition with food and other sustainability concerns, which could be addressed by resorting to marginal land (Rogelj et al. (2016) and Creutzig et al. (2015)).

Notably, in our model, we used conservative estimates for both biomass residues and marginal land available, which we have failed to clarify in the original manuscript. **In the new revised version, we have better explained how we computed our biomass estimates and the assumptions behind our calculations (see Supplementary Information pages S40 and S41, lines 739-758 and page S41, lines 764-768).** In essence, we consider the potentials of agricultural and forestry residues based on sustainability criteria. The latter include assumptions concerning the biomass left for soil conservation and biodiversity protection and other competitive uses of such residues (e.g., straw for animal bedding or pruning for firewood). Regarding the marginal land availability estimates, we followed the same procedure as in a recent work (Pozo et al. (2020)). Notably, we consider the conservative scenario from Cai et al. (2010), which includes at least part of the abandoned, wasted, or idle agricultural land and some small crop fields. Then, we adjusted the marginal land availability by downgrading the original estimates by 69%, according to Fritz et al. (2013). Overall, we found that our marginal land availability estimations are lower than others recently published [1] (e.g., in [1], the marginal land area in Poland is 25,796 km² while our estimate is 7,371 km², thus more conservative).

Nevertheless, acknowledging that the marginal land and biomass residues availability are uncertain, we have now performed a sensitivity analysis on these parameters as the Reviewer suggested. The details on the sensitivity analysis calculations can be found in Methods (page 20, lines 557-562), while the results are given in the Supplementary Information document (Supplementary Fig. 2 in page S49, shown below for convenience). Moreover, we have discussed briefly in the main text the effect of uncertainties in the biomass resources available on the outcome of the optimization model (see pages 5 and 6, lines 132-145).

Rogelj, J., Den Elzen, M., Höhne, N., Fransen, T., Fekete, H., Winkler, H., ... & Meinshausen, M. (2016). Paris Agreement climate proposals need a boost to keep warming well below 2 C. Nature, 534(7609), 631-639.

Creutzig, F. et al. Bioenergy and climate change mitigation: an assessment. GCB Bioenergy 7, 916–944 (2015)

Pozo, C., Galán-Martín, Á., Reiner, D. M., Mac Dowell, N., & Guillén-Gosálbez, G. (2020). Equity in allocating carbon dioxide removal quotas. Nature Climate Change, 10(7), 640-646.

Cai, X., Zhang, X. & Wang, D. Land availability for biofuel production. Environ. Sci. Technol. 45, 334–339 (2010).

Fritz, S. et al. Downgrading recent estimates of land available for biofuel production. Environ. Sci. Technol. 47, 1688–1694 (2013).

[1] Marginal lands for Growing Industrial Crops: Turning a burden into an opportunity.
<https://cordis.europa.eu/project/id/727698/results/es>
<https://www.arcgis.com/apps/webappviewer/index.html?id=a813940c9ac14c298238c1742dd9dd3c>

Supplementary Fig. 2. Sensitivity analysis on the biomass potentials (biomass residues and marginal land). Dots correspond to the optimal solutions deploying BECCS and DACCS from a particular point in time onwards (from 2020 to 2100). The shaded areas indicate the new results for a given percentage change in biomass potentials. Subplot **a** shows the sensitivity analysis for the minimum costs of the EU power system associated with increasing CDR targets. Subplot **b** corresponds to the sensitivity analysis for the maximum CDR attainable. In subplot **b**, the green

profile considers only BECCS, blue DACCS, and yellow both BECCS and DACCS, while the pie charts illustrate the proportion of gross CDR provided with BECCS and DACCS, respectively.

Comment 3: *Region Minor remarks:*

Introduction

74| I would recommend to add “to achieve a net CO₂ emissions target in the power sector” as other sector’s emissions are not fully covered in the model used in this article.

Authors: Thanks for the comment. **We have clarified that the target is defined on the life-cycle emissions of the power sector (page 3, line 75). Besides implementing the amendment mentioned by the Reviewer, we have also clarified this point throughout the text (e.g., page 5, line 128).**

Comment 4: *Results and discussion*

Fig 1b) just for clarification: Why is BECCS deployment not restricted by diffusion rates, while DACCS deployment is critically limited in the delayed cases?

Authors: Thanks for this interesting question. The reason is that, in the case of BECCS, the unused biomass resources (residues and marginal land) are the main factor constraining BECCS deployment when there are (1) enough remaining years until 2100 and (2) sufficient storage capacity to store the biogenic CO₂ captured. Hence, CDR inaction entails that biomass residues are not mobilized, and marginal land is not exploited, leading to a loss of biomass potential decreasing linearly with time. However, after a point in time (i.e., 2090), the slope of the curve significantly decreases due to the maximum diffusion constraint, which becomes the bottleneck at a certain point (see Fig.1b green curve). The maximum attainable capacity dictated by this constraint strongly depends on the initial installed capacity of the technology (Supplementary Fig. 4 in revised version), which is much smaller for DACCS: 1 Mton/yr, reflecting the current ambition of the Carbon Engineering plant in Texas (still under construction), and 250 MW for BECCS in each of the 28 EU countries (based on the largest existing biomass power plants, which results in 7,000 MW at the European level). Therefore, the different installed capacities at time zero and the fact that BECCS is also constrained by biomass and land availability explain the different shapes of the potential curves.

Comment 5: *152-154| DACCS diffusion rate and initial capacity are given. Could you add your BECCS initial capacity assumptions as well?*

Authors: Yes, thanks for the suggestion. We selected an initial capacity for BECCS of 250MW, reflecting the state-of-the-art largest standalone biomass-fired combustion power plant. This information was reported in the Supporting Information (page S46, lines 815-817). **We have further clarified this point in the main text accordingly (page 7, lines 178-180).**

Comment 5: *Regional implications for the energy systems*

Fig 3a) Having such a large share of nuclear in the German power mix is extremely unlikely, but probably necessary in the RAPID model in the absence of storage technologies. I don’t expect the

fundamental results concerning CDR deployment to change much with different power system technologies available, but it would be good to mention alternatives presented in the literature. (e.g. Victoria et al. 2020, Gerbaulet et al. 2020, Plessmann et al. 2017, Pietzcker et al. 2021)

Authors: Thanks for the comment. Note that in our RAPID model (see Eq. 13 in the Supplementary Information), the capacity installed for nuclear power is constrained by a limit. This upper bound corresponds to today's installed capacity, implying that we cannot expand the current nuclear power capacity. This assumption is based on the recent emergence of phase-out plans for nuclear power in Europe (e.g., Germany, Belgium, or Switzerland). The same assumption was also adopted in some of the works suggested by the Reviewer (Plessmann et al. (2017) and Pietzcker et al. (2021)). **We have clarified this point in the Supplementary Information (page S41, lines 754-758) while adding the references suggested by the Reviewer to support our modeling approach further.**

Moreover, according to our results (pie charts in Fig. 3a showing the power technologies shares by 2100), the nuclear share in Germany increases (from ca. 12% in 2020 to ca. 23% by 2100), yet the nuclear capacity would remain the same. Note that Germany imports electricity from abroad (i.e., from France and Sweden, see Fig 4c), thus reducing the amount of electricity generated domestically. **We have also clarified in the text that no new nuclear capacity is installed as capacity expansions are not allowed (page 12, line 317-318).** Hence, our results are fully aligned with the works suggested by the reviewer (e.g., Victoria et al. 2020 and Gerbaulet et al. 2020). **We have also added a reference to the article by Victoria et al. (2020) since we found it highly relevant for our work.**

Victoria, M., Zhu, K., Brown, T., Andresen, G. B., & Greiner, M. (2020). Early decarbonisation of the European energy system pays off. Nature communications, 11(1), 1-9.

Gerbaulet, C., von Hirschhausen, C., Kemfert, C., Lorenz, C., & Oei, P. Y. (2019). European electricity sector decarbonization under different levels of foresight. Renewable energy, 141, 973-987.

Plessmann, G., & Blechinger, P. (2017). How to meet EU GHG emission reduction targets? A model based decarbonization pathway for Europe's electricity supply system until 2050. Energy Strategy Reviews, 15, 19-32.

Pietzcker, R. C., Osorio, S., & Rodrigues, R. (2021). Tightening EU ETS targets in line with the European Green Deal: Impacts on the decarbonization of the EU power sector. Applied Energy, 293, 116914.

Comment 6: Supplemental Material

132 | "back-up" typo

466 | Table 10, Geothermal 2020(p1) typo

821 – 824 | Table 47 and Table 48 are identical

Authors: Thanks for spotting these typos. **We have amended them accordingly.**

Comment 7: Supplementary Fig. 2 | It would be nice to have a dotted version of BECCS using the Maximum achievable carbon removal plotted to the sequestration axis together with DACCS.

Authors: We thank the Reviewer for this comment regarding Supplementary Fig. 2 (now renumbered as Supplementary Fig. 4 after the new additions). The Reviewer's suggestion is very interesting, yet plotting such a curve poses some methodological challenges, not least because the BECCS is a power technology whose capacity is often expressed in terms of power while that of DACCS typically refers to its gross CO₂ removal potential. Expressing the former in terms of CO₂ removed requires some additional assumptions that we would like to avoid to prevent the potential misinterpretation of the figure. Notably, the functional unit for the DACCS technology is the amount of gross CO₂ removed, which is independent of the power source and location. In contrast, the gross CO₂ removed by BECCS is given by the CO₂ uptake via photosynthesis during biomass growth, which in turn depends on the biomass type and location. Hence it would not be possible to express the capacity of BECCS facilities in terms of gross CO₂ removed without assuming a biomass type and location. Moreover, note the net carbon efficiency of the BECCS (i.e., net CO₂ removed / CO₂ gross CO₂ removed) is highly case-specific and varies greatly depending on the life cycle emissions taking place in each particular supply chain. For example, the BECCS removal capacity would be different for Miscanthus or prunings. Consequently, it is more sensible to impose the maximum diffusion rate on the power capacity installed, similarly as done with the remaining power technologies considered in our study.

Reviewer comments, further round review -

Reviewer #1 (Remarks to the Author):

1. This piece provides a lot of value-added in the realm of CDR policymaking, not only in terms of the implications of the timing of deployment of CDR options, but also granular issues of how to divide up potential responsibilities in the European context in terms of questions e.g. sourcing bioenergy feedstocks and storage;
2. Line 53 on p.2: Why is the word "hence" used at the outset of the sentence? The argument here doesn't follow from the text immediately before it;
3. I would have liked a bit more discussion of the new EU Climate Law and its implications for your estimates of optimal deployment of BECCS/DACCS given the limits that it places on deployment of CDR as part of the overall carbon budget;
4. At this point, I don't think anyone conceives of "delaying" deployment of CDR options, but the question is how quickly do we scale up its utilization;
5. While you conclude that large-scale BECCS can be sustainably deployed with marginal land, I'm not sure this is the case in terms of fertilizer use, which is even more intensive on marginal land. See the new piece by Li in Environmental Science & Technology, which concludes that fertilizer demand for large-scale BECCS renders it an unsustainable option;
6. I think the "crowding-out" effect of CCS and fossil fuels in terms of carbon storage may be valid, but I'm not so sure that storage capacity in the EU is as constrained;
7. I thought that my previous concerns in this piece were well-addressed and that it would constitute a valuable contribution in the field.

Reviewer #2 (Remarks to the Author):

Thank you very much for your kind, insightful and comprehensive reply to the review points. I have nothing else to add.

Referees' comments in blue - Replies in black - Actions in bold . Indicated page and line refer to the revised manuscript and ESI.

Response to Comments from Reviewer #1

Comment 1: *1. This piece provides a lot of value-added in the realm of CDR policymaking, not only in terms of the implications of the timing of deployment of CDR options, but also granular issues of how to divide up potential responsibilities in the European context in terms of questions e.g. sourcing bioenergy feedstocks and storage;*

Authors: We thank the Reviewer for this positive comment. Besides raising concerns on the risk of inaction on CDR, we fully agree that our results also provide valuable insights on how to optimally deploy BECCS and DACCS in the context of the EU power system. Hence, **we have added a short sentence highlighting that our results could help shape national CDR roadmaps and policies to promote full-scale BECCS and DACCS projects in the EU (page 16, lines 414-415).**

Comment 2: *2. Line 53 on p.2: Why is the word "hence" used at the outset of the sentence? The argument here doesn't follow from the text immediately before it;*

Authors: Thanks for this comment. Indeed, we agree that both sentences refer to different arguments. **Accordingly, we have replaced "Hence" with "Moreover" (see page 2, lines 58).**

Comment 3: *3. I would have liked a bit more discussion of the new EU Climate Law and its implications for your estimates of optimal deployment of BECCS/DACCS given the limits that it places on deployment of CDR as part of the overall carbon budget;*

Authors: We thank the Reviewer for this comment, which was indeed addressed to some extent in the previous iteration. We selected the EU for the analysis due to its expected role in climate change mitigation, as outlined in the EU Green Deal and its EU Climate Law. Besides providing relevant insights for the EU policymaking context, our findings could be extrapolated to any region/country or even at the global level. For this reason, we decided to explore the implications for emissions and costs for different removal targets (not focusing only on carbon-neutrality by 2050, as stated in the law above). Furthermore, regulations and targets tend to be reviewed and amended when goals are not being met, so while current targets are useful for a specific context, we would prefer not to be overly constrained by their current level of ambition. The new EU Climate Law recognizes different roles for CO₂ reduction and removal. However, it only explicitly mentions removals from land sinks (land-use, land-use change, and forestry measures), establishing maximum contribution levels from them to reach the interim 55% reduction target by 2030. Hence, it fails to mention technological sinks such as BECCS or DACCS explicitly. Nevertheless, to reach climate neutrality by 2050 in the EU as a whole (and after 2050), some EU countries and/or sectors will need to go beyond carbon neutrality and achieve net-negative emissions. We argue that the EU power sector will play a key role in providing those negative emissions needed to offset very hard to abate emissions (e.g., methane from livestock and aviation). Therefore, promoting the near-term integration of those technologies at the

earliest is crucial to ensure that they can be deployed to the extent required to meet the long-term goals. **We have added this discussion in the main manuscript (page 7, lines 190-200).** Note also that, in the previous round of revision, we added additional results for 2050 to highlight the relevance of inaction on CDR for the EU climate neutrality strategy. We found that delaying CDR actions beyond 2040 might prevent the EU power sector from reaching carbon neutrality in 2050 (e.g., -0.85 Gt CO₂ by 2050 starting in 2040) while increasing the costs in the range of 0.04–0.10 trillion EUR per year of inaction (Supplementary Fig. 3). Due to space limitations, we provided these new results in the Supplementary Information (Supplementary Fig. 3), discussing them briefly in the main manuscript.

Comment 4: *4. At this point, I don't think anyone conceives of "delaying" deployment of CDR options, but the question is how quickly do we scale up its utilization;*

Authors: We thank the Reviewer for this comment. We think that both statements are somehow connected, as delaying CDR, understanding by delaying not full inaction but rather lack of strong action, will slow the scale-up of these technologies (particularly those with low technology readiness levels). Neither BECCS nor DACCS value chains have been deployed at scale yet, and are still at the early stages of demonstration. Moreover, CDR incentives are currently scarce, although scaling up these technologies will pose technical challenges already identified by Integrated Assessment Models.

Comment 5: *5. While you conclude that large-scale BECCS can be sustainably deployed with marginal land, I'm not sure this is the case in terms of fertilizer use, which is even more intensive on marginal land. See the new piece by Li in Environmental Science & Technology, which concludes that fertilizer demand for large-scale BECCS renders it an unsustainable*

Authors: We thank the Reviewer for providing this interesting article, which was out of our radar. **We have added a short discussion on the sustainability concerns associated with the fertilizers required to sustain the massive deployment of BECCS, citing the reference provided accordingly (see page 5, lines 138-140).**

Comment 6: *6. I think the "crowding-out" effect of CCS and fossil fuels in terms of carbon storage may be valid, but I'm not so sure that storage capacity in the EU is as constrained;*

Authors: We thank the Reviewer for this comment. As mentioned in our work, our estimates only consider onshore geological sites, although the EU domestic storage capacity could be increased by including offshore storage sites (see page 6, lines 151-156). The CO₂ geological capacity in Europe was retrieved from the Geocapacity project [1], regarded as the most reliable and conservative source for sedimentary basins suitable for the geological storage of CO₂ in the EU countries. Notably, these estimates are not theoretical (which could lead to unrealistic capacities) but rather consider technical constraints and economic and regulatory barriers whenever available. Nevertheless, further research will be needed to ultimately define the suitability of each specific storage site based on a full range of technical, economic, and environmental constraints. **This point has been further clarified in the assumptions and limitations of the work.**

[1] Vangkilde-Pedersen, T. et al. Assessing European capacity for geological storage of carbon dioxide—the EU GeoCapacity project. Energy Procedia 1, 2663–2670 (2009).

Comment 7: *7. I thought that my previous concerns in this piece were well-addressed and that it would constitute a valuable contribution in the field.*

Authors: We are glad that the previous Reviewer’s concerns were well addressed. We also think that our manuscript is timely and could be valuable for the community. We greatly appreciate the Reviewer’s constructive comments that definitely helped improve the manuscript.